# DEFINING AND MEASURING DISENTANGLEMENT FOR NON-INDEPENDENT FACTORS OF VARIATION

## ABSTRACT

Representation learning is an approach that allows to discover and extract the factors of variation from the data. Intuitively, a representation is said to be disentangled if it separates the different factors of variation in a way that is understandable to humans. Definitions of disentanglement and metrics to measure it usually assume that the factors of variation are independent of each other. However, this is generally false in the real world, which limits the use of these definitions and metrics to very specific and unrealistic scenarios. In this paper we give a definition of disentanglement based on information theory that is also valid when the factors are not independent. Furthermore, we demonstrate that this definition is equivalent to having a representation composed of minimal and sufficient variables. Finally, we propose a method to measure the degree of disentanglement from the given definition that works when the factors are not independent. We show through different experiments that the method proposed in this paper correctly measures disentanglement with independent and non-independent factors, while other methods fail in the latter scenario.

## 1 INTRODUCTION

In the jargon of representation learning, data are considered to be completely defined by factors of variation (or simply factors from now on) and nuisances. The difference between them is that the former are relevant for a given task and the latter are not. For example, if we had a dataset composed of images of fruits and the task were to describe the fruits in the images, the factors could be the type of fruit, the color or the size. On the other hand, nuisances could be the background color or the shadow of the fruit. In multiple works it has been proposed that it is desirable for a representation to be disentangled, i.e., it separates the different factors of variation (Schmidhuber, 1992; Bengio et al., 2013; Ridgeway, 2016; Lake et al., 2017; Achille & Soatto, 2018a). Disentangled representations have different application, such as: (i) interpreting and explaining the representations and predictions (Hsu et al., 2017; Worrall et al., 2017; Gilpin et al., 2018; Zhang & Zhu, 2018; Liu et al., 2020; Zhu et al., 2021; Nauta et al., 2023; Almudévar et al., 2024), (ii) making fairer predictions by mitigating or eliminating biases in sensitive attributes (such as gender or race) (Calders & Žliobaitė, 2013; Barocas & Selbst, 2016; Kumar et al., 2017; Locatello et al., 2019a; Sarhan et al., 2020; Chen et al., 2023), or (iii) giving generative models the ability to create new data with concrete attributes (Yan et al., 2016; Paige et al., 2017; Huang et al., 2018; Dupont, 2018; Kazemi et al., 2019; Antoran & Miguel, 2019; Zhou et al., 2020; Shen et al., 2022; Almudévar et al., 2024).

The definition in the previous paragraph allows us to understand the concept of disentangled representation in an intuitive but imprecise way. Two main questions arise from this definition: what does it exactly mean to separate the factors, and how can we assess whether the different factors are actually separated? We review in section 2 different works that have tried to clear these doubts. Most of these works assume that the different factors are independent of each other and of the nuisances. Therefore, the definitions and metrics they propose are assuming this independence. Most of the datasets where these definitions and metrics are assessed do have their factors independent (or nearly independent) (LeCun et al., 2004; Liu et al., 2015; Reed et al., 2015; Matthey et al., 2017; Burgess & Kim, 2018; Gondal et al., 2019), so the definitions and metrics are valid (or nearly valid).

However, in the real world it is uncommon for factors to be independent of each other and of the nuisances. In the previous example, it is easy to see that the type of fruit is not independent of the

color, the size or the shadow of the fruit. It is very likely to have a red strawberry, but the probability of having a red banana tends to zero. When the factors are independent of each other, understanding the concept of separating them seems obvious. However, it is not so obvious to understand what it means for a representation to separate dependent factors. For example, we might ask whether it is possible to separate the concept of a strawberry from the color red and what the implications of this are. If we can separate the factors, we could create an image of a red banana or avoid classifying a yellow strawberry as a banana or a lemon just for the fact that it is yellow, for example. In this work, we focus on defining and measuring disentanglement when the factors are not independent of each other and of the nuisances. Our key contributions can be summarized as follows:

- We propose a set of four properties based on information theory that are desirable for a disentangled representation when the factors are not independent of each other and of the nuisances. We connect these properties to those most accepted in the literature.

- We demonstrate that these properties are fulfilled if and only if the representations are minimal and sufficient. Then, we argue that measuring minimality and sufficiency is more interesting than measuring the previous properties separately.

- We give a method for measuring the degree of minimality and sufficiency, which allows to evaluate the level of disentanglement of a representation. We evaluated these methods on different datasets to illustrate their appropriateness in a variety of scenarios.

## 2  RELATED WORK

**Definition of Disentanglement**  Despite being a topic of great interest, there is no general consensus on the definition of disentangled representation. Intuitively, a disentangled representation separates the different factors of the data (Desjardins et al., 2012; Bengio et al., 2013; Cohen & Welling, 2014; Kulkarni et al., 2015; Ridgeway, 2016). Originally, disentanglement was evaluated via visual inspection (Kingma & Welling, 2013). However, there is a need to better specify what it means for a representation to be disentangled, as this provides insight into the scope and limitations of their different applications. One of the first definitions of disentanglement was given in Bengio et al. (2013) and long the most widely accepted in the literature (Higgins et al., 2017; Kim & Mnih, 2018; Kim et al., 2019; Suter et al., 2019), says that a disentangled representation is one in which a change in one part of the representation corresponds to a change in a factor of variation, while being relatively invariant to changes in other factors. Other authors claim that a disentangled representation is one in which a change in a single factor of variation translates into the change in a single part of the representation (Kumar et al., 2017; Locatello et al., 2019b). Others propose to define a disentangled representations based on a list of properties that they must satisfy. Ridgeway & Mozer (2018) and Eastwood & Williams (2018) stand out in this by proposing virtually simultaneously three properties that a representation must satisfy in order to be considered disentangled. Although they refer to the same ideas, the two papers refer to these properties by different names. These properties are: (i) modularity (disentanglement in Eastwood & Williams (2018)), that is, each variable in the representation captures at most one factor of variation; (ii) compactness (completeness in Eastwood & Williams (2018)), i.e., a factor of variation is captured by only one variable of the representation; and (iii) explicitness (informativeness in Eastwood & Williams (2018)), that is, the representation captures all the information about the factors that is present in the input. All of the above definitions present two problems: (i) they do not analyze the invariance to some nuisances possibly present in the input; and (ii) they assume that the factors of variation are independent of each other and of the nuisances, which is in general false. In this paper we give a list of properties that a representation must satisfy to be considered disentangled considering these two last aspects.

**Metrics for Disentanglement**  As a consequence of the fact that there is no consensus on the definition of disentanglement, there is no consensus on how to measure it. In Carbonneau et al. (2022) they organize the methods in the literature into three groups according to the principle of their operation. Below we explain an overview of each of these groups.

- **Intervention-based Metrics.** These create subsets in which their elements have in common a factor of variation while the rest are different. Subsequently, the representations of these elements are obtained and compared in different ways to obtain a score. Some of the methods in this family are: $\beta$-VAE (Higgins et al., 2017), FactorVAE (Kim & Mnih, 2018) and R-FactorVAE (Kim et al., 2019). These methods are intended to measure only modularity, but not compactness and explicitness (Sepliarskaia et al., 2019).

- **Information-based Metrics.** These methods estimate the mutual information between the factors of variation and each variable in the representation. Some of the most widely used are Mutual information Gap (MIG) (Chen et al., 2018) or Robust MIG (Do & Tran, 2019). These methods are designed to measure only the compactness. Modifications on these methods have been proposed to also capture modularity (Do & Tran, 2019; Li et al., 2020; Ridgeway & Mozer, 2018; Sepliarskaia et al., 2019).

- **Predictor-based Metrics.** These train regressors or classifiers to predict the factors of variation from the different variables in the representations. Subsequently, the predictor is inspected to analyze the importance of each variable to predict each factor. The first method of this family proposed was Separated Attribute Predictability (SAP) (Kumar et al., 2017), which measures only compactness. Soon after, as we have already explained, it was proposed to measure disentanglement through different properties. In Eastwood & Williams (2018) the properties disentanglement, completeness and informativeness (DCI) and ways to measure them are proposed. Simultaneously, the properties modularity, compactness and explicitness and ways to measure them are proposed in Ridgeway & Mozer (2018).

Our proposal would fall into the latter group, since we train predictors to measure the different properties we describe. However, unlike the methods mentioned above, ours allows us to measure the degree of disentanglement even when the factors are dependent between them or of the nuisances.

**Disentanglement under Dependent Factors of Variation** As mentioned, disentanglement under dependent factors is an underexplored field. However, there are some works related to this idea, among which the following stand out: Suter et al. (2019), that gives a causal perspective on representation learning that addresses disentanglement and domain shift robustness; Choi et al. (2020), which disentangles variations common to all classes and variations exclusive of each class; Montero et al. (2020; 2022), which excludes combinations of factors during training and at test time they ask the models to reconstruct the missing; Träuble et al. (2021), that artificially introduces dependencies between pairs of factors; Roth et al. (2022), which considers the use of Hausdorff Factorized Support (HFS) criterion allowing for arbitrary distributions of the factors over their support, including correlations between them; Ahuja et al. (2023), that explores how interventional data can help causal representation learning by revealing geometric patterns in latent variables; and von Kügelgen et al. (2024), which focuses on inferring latent causal variables and relations from high-dimensional data using a non-parametric approach without restrictive assumptions.

**Information Bottleneck** This a technique introduced by Tishby et al. (2000) that is designed to find a trade-off between fidelity and compression of representations. The representation that is maximally faithful is said to be sufficient and the one that is maximally compressed is called minimal. The ideal representation should contain only (minimal) all (sufficient) the information necessary to solve a given task. Formally, given an input $x$, a task $y$ and a representation $z$, the sufficient representation $z$ maximizes $I(z; y)$ and the minimal representation minimizes $I(z; x)$. Different works use this technique to derive regularizers in representation learning (Alemi et al., 2016; 2018; Achille & Soatto, 2018b;a). Others have optimized it to obtain disentangled representations. Vera et al. (2018); Yamada et al. (2020); Jeon et al. (2021); Gao et al. (2021). However, to the best of our knowledge, the present is the first work formally defining the concept of disentanglement through that of minimal sufficient representation. We do it in a way in which we consider each of the factors as a task and the representation as a set of representations. Concretely, if each of these representations is minimal and sufficient for a given factor, then we consider the representation to be disentangled.

## 3 DESIRABLE PROPERTIES OF A DISENTANGLED REPRESENTATION

In our representation learning task we have a raw data $x$ which is fully explained by a set of factors of that raw data $y = \{y_i\}_{i=1}^n$ and a set of nuisances $n$. The factors $y$ refer to the underlying sources or causes that influence the observed data and that are relevant to a given task. We assume that these factors are conditionally independent of each other given $x$, i.e., $p(y_i, y_j | x) = p(y_i | x) p(y_j | x)$ for $i \neq j$ [1]. The nuisances $n$ refer to the variations present in $x$ that are irrelevant to a given task.

A representation $z$ is a variable that is fully described by the distribution $p(z | x)$. Therefore, we have the Markov chains $y \leftrightarrow x \leftrightarrow z$ and $y_i \leftrightarrow x \leftrightarrow z$ for $i = 1, 2, \ldots, n$. The goal of representation

---

[1] Most of the works in the literature consider $y_i$ and $y_j$ to be independent, that is, $p(y_i, y_j) = p(y_i)p(y_j)$ for $i \neq j$, but this is not true in most real world scenarios.

learning is to obtain representations of the inputs. A representation $\boldsymbol{z}$ can be viewed as a set of representations such that $\boldsymbol{z} = \{z_j\}_{j=1}^m$, which are also fully described by $p(z_j|\boldsymbol{x})$. From now on we will refer to the $z_j$ as the variables of $\boldsymbol{z}$. In some works they restrict the $z_j$ to scalars. However, in this paper we do not consider this restriction, since it is in general impossible to fully describe a factor of variation by using an scalar.

Next, we describe four different properties that are desirable for a disentangled representation. They are connected to those described in section 2. However, the latter do not consider the case where the factors are dependent and the ones we propose do. In fact, we could view three of these properties as an adaptation of those defined in Ridgeway & Mozer (2018) and Eastwood & Williams (2018) for the case in which the factors are not independent between them and of the nuisances. For the definitions of these properties, we assume that $z_j$ is intended to describe $y_i$.

- **Factors-Invariance**: We say that a variable $z_j$ of the representation $\boldsymbol{z}$ is factors-invariant for a factor $y_i$ when it satisfies the next Markov Chain:

$$\tilde{\boldsymbol{y}}_i \leftrightarrow y_i \leftrightarrow z_j \tag{1}$$

where $\tilde{\boldsymbol{y}}_i = \{y_k\}_{k \neq i}$. This is equivalent to having $I(z_j; \tilde{\boldsymbol{y}}_i|y_i) = 0$, i.e., once $y_i$ is known, $z_j$ will be the same regardless of all other factors. This property is directly connected to modularity (Ridgeway & Mozer, 2018) (or disentanglement in Eastwood & Williams (2018)): a representation $z_j$ is said to be modular (or disentangled) if it captures at most one factor $y_i$. However, this last definition is convenient only when the factors are independent of each other. Imagine that we have two factors $y_i$ and $y_k$ dependent on each other, then, according to this definition, $z_j$ could never be modular and store all the information about $y_i$ at the same time, since $y_k$ contains information about $y_i$.

- **Nuisances-Invariance**: We call a variable $z_j$ of the representation $\boldsymbol{z}$ nuisances-invariant for a factor $y_i$ and nuisances $\boldsymbol{n}$ when it satisfies the next Markov Chain:

$$\boldsymbol{n} \leftrightarrow y_i \leftrightarrow z_j \tag{2}$$

This is equivalent to having $I(z_j; \boldsymbol{n}|y_i) = 0$, i.e., once $y_i$ is known, $z_j$ will be the same regardless of the nuisances. This property is typically neither mentioned nor measured, which could be due to the difficulty of its measurement. This difficulty comes from the fact that it is complicated to estimate the distribution of $\boldsymbol{n}$. However, it is a relevant property for a disentangled representation. Through factors-invariance we are measuring that all factors other than $y_i$ do not affect $z_j$, but in a disentangled representation we would expect that everything that is part of the input and is not a factor of variation (i.e. a nuisance) does not affect $z_j$ either (Carbonneau et al., 2022).

- **Representations-Invariance**: We call a variable $z_j$ of the representation $\boldsymbol{z}$ representations-invariant for a factor $y_i$ when it satisfies the next Markov Chain:

$$\tilde{\boldsymbol{z}}_j \leftrightarrow z_j \leftrightarrow y_i \tag{3}$$

where $\tilde{\boldsymbol{z}}_j = \{z_k\}_{k \neq j}$. This is equivalent to having $I(y_i; \tilde{\boldsymbol{z}}_j|z_j) = 0$, i.e., $y_i$ does not need any variable of $\boldsymbol{z}$ other than $z_j$ to be defined. This can be useful for downstream tasks: if we want to predict $y_i$, we could keep only $z_j$ and ignore the rest. Also, in a controllable generative model, it would be sufficient to manipulate only $z_j$ to determine the value of $y_i$ in $\boldsymbol{x}$. This property is closely connected to compactness (Ridgeway & Mozer, 2018) (or completeness in Eastwood & Williams (2018)): a representation $z_j$ is said to be compact (or complete) if it is the only one that captures a factor $y_i$. For the same reason as in factors invariance, this last definition is only convenient if the factors are independent of each other. Let $y_i$ and $y_k$ be two dependent factors represented by $z_j$ and $z_l$, respectively. Then it is expected that both $z_j$ and $z_l$ have information about both $y_i$ and $y_k$.

- **Explicitness** We call a representation $\boldsymbol{z}$ explicit for a factor $y_i$ when it satisfies the next Markov Chain:

$$\boldsymbol{x} \leftrightarrow \boldsymbol{z} \leftrightarrow y_i \tag{4}$$

This means that $I(y_i; \boldsymbol{x}|\boldsymbol{z}) = 0$, i.e., $\boldsymbol{x}$ provides no information about $y_i$ when $\boldsymbol{z}$ is known or, equivalently, $\boldsymbol{z}$ contains all the information about $y_i$. This property is defined in Ridgeway & Mozer (2018) with the name explicitness or in Eastwood & Williams (2018) with the name informativeness and it is not desirable only for a disentangled representation but for a representation in general (Bengio et al., 2013).

## 4 Disentanglement through Minimality and Sufficiency

In representation learning, given an input $x$ and a task $y$, the representation $z$ that maximizes $I(z; y)$ is called sufficient. By the definition of representation, we know by the Data Processing Inequality (DPI) (Beaudry & Renner, 2011) that $I(z; y) \leq I(x; y)$. Since this is the unique upper bound of $I(z; y)$, the sufficient representation satisfies that $I(z; y) = I(x; y)$ or, equivalently, the Markov chain $x \leftrightarrow z \leftrightarrow y$, i.e., $y$ is fully described by $z$. On the other hand, the representation $z$ that minimizes $I(z; x)$ is called minimal. From the definition of representation and the DPI, we know that $I(z; x) \geq I(z; y)$. Since this is the unique lower bound of $I(z; x)$, the minimal representation satisfies that $I(z; x) = I(z; y)$ or, equivalently, the Markov chain $x \leftrightarrow y \leftrightarrow z$, i.e., $z$ is fully described by $y$. From the above, we have that a representation is sufficient and minimal if it satisfies that $I(x; y) = I(z; y) = I(z; x)$.

### 4.1 Connection between Disentanglement and Minimality and Sufficiency

**Proposition 1.** *Let $\boldsymbol{y} = \{y_l\}_{l=1}^n$ be some factors and $\boldsymbol{z} = \{z_j\}_{j=1}^m$ a representation. Then, $z_j$ is a minimal representation of $y_i$ if and only if $z_j$ is factors-invariant for $y_i$ and nuisances-invariant. Equivalently, we have that:* [2]

$$(\boldsymbol{x} \leftrightarrow y_i \leftrightarrow z_j) \Longleftrightarrow (\tilde{\boldsymbol{y}}_i \leftrightarrow y_i \leftrightarrow z_j) \wedge (\boldsymbol{n} \leftrightarrow y_i \leftrightarrow z_j)$$

where $\tilde{\boldsymbol{y}}_i = \{y_l\}_{l \neq i}$. Therefore, satisfying minimality is equivalent to satisfying factors-invariance and nuisances-invariance jointly. Intuitively, if $z_j$ is completely defined by $y_i$ (minimal), then neither the rest of the factors of $\boldsymbol{y}$ nor the nuisances $\boldsymbol{n}$ affect $z_j$ once $y_i$ is known (factors-invariance and nuisances-invariance, respectively). Similarly, since $\boldsymbol{x}$ is completely defined by $\boldsymbol{y}$ and $\boldsymbol{n}$, the reciprocal is also true. We prove this proposition in Appendix A.

**Proposition 2.** *Let $y_i$ be a factor and $\boldsymbol{z} = \{z_l\}_{l=1}^m$ a representation. Then, $z_j$ is a sufficient representation of $y_i$ if and only if $z_j$ is representations-invariant for $y_i$ and $\boldsymbol{z}$ is explicit for $y_i$. Equivalently, we have that:*

$$(\boldsymbol{x} \leftrightarrow z_j \leftrightarrow y_i) \Longleftrightarrow (\tilde{\boldsymbol{z}}_j \leftrightarrow z_j \leftrightarrow y_i) \wedge (\boldsymbol{x} \leftrightarrow \boldsymbol{z} \leftrightarrow y_i)$$

where $\tilde{\boldsymbol{z}}_j = \{z_l\}_{l \neq j}$. Thus, satisfying sufficiency is equivalent to satisfying representations-invariant and explicitness jointly. Intuitively, if $y_i$ is completely defined by $z_j$ regardless $\boldsymbol{x}$ (sufficiency), then there is no information about $y_i$ that is in $\boldsymbol{z}$ and is not in $z_j$ (representations-invariance) and $y_i$ is completely defined by $\boldsymbol{z}$ (explicitness) since $z_j \in \boldsymbol{z}$. Similarly, if all information present in $\boldsymbol{z}$ about $y_i$ is present in $z_j$ (representations-invariance) and $\boldsymbol{z}$ captures all $y_i$ (explicitness), then $z_j$ fully describes $y_i$ (sufficiency). We prove this proposition in Appendix A.

### 4.2 Why measuring Disentanglement through Minimality and Sufficiency?

In section 3 we have provided four desirable properties for a disentangled representation. These properties hinge on those most widely accepted in the literature and are adapted to the scenario in which the factors may be non-independent. In section 4.1 we connect this properties to the concepts of minimality and sufficiency. Next, we argue that it makes more sense to evaluate minimality and sufficiency than the four properties of 3 separately to measure the degree of disentanglement.

1. **Minimality vs. Factors-Invariance + Nuisance Invariance**: When we evaluate whether a representation is disentangled, we really want to analyze whether each of its variables is affected only by a single factor regardless of the other factors or the nuisances as a whole. It is no use having a variable that is only affected by one of the factors if the nuisances affect it to a large extent. Similarly, if a variable is nuisances-invariant, but the rest of the factors affect it, we could not consider that we have an disentangled representation. Therefore, we argue that it makes sense to measure these two properties jointly and, as proposed in Proposition 1, this can be done through measuring the minimality.

2. **Sufficiency vs. Representation-Invariance + Explicitness**: When we evaluate whether a representation is disentangled, we want to analyze if we can describe a factor using only one variable of the representation and whether we can describe it completely. If we had a representation in which only one variable affected a factor, but could describe it to a very low extent, it would not be fulfilling the most elementary objective of a representation,

---

[2] Do not confuse $\Longleftrightarrow$, which means if and only if, and $\leftrightarrow$, which is an arrow of the Markov chain.

which is to describe the factors. Similarly, if a factor were well defined by a representation, but all variables affected this factor equally, then we could not say that the representation is disentangled. Therefore, we believe that measuring these two properties jointly is more convenient than separately. As proposed in Proposition 2, this can be done through measuring the sufficiency.

Due to the aforementioned, we propose metrics to measure the degree of minimality and sufficiency of the variables of $z$ and, hence, its degree of disentanglement. Notwithstanding the foregoing, it could be that one would be interested in measuring also the different properties of 3 individually. Thus, in Appendix C we propose methods to measure all these properties separately except nuisances-invariance because of the intrinsic difficulty (or impossibility) of estimating the distribution of the nuisances. In this appendix we also give some examples that illustrate the convenience of using sufficiency and minimality to evaluate the degree of disentanglement compared to measuring properties in section 3 separately.

### 4.3 METRICS FOR MINIMALITY AND SUFFICIENCY IN PRACTICE

As we have explained, a variable $z_j$ is minimal with respect to $y_i$ when $I(z_j; \boldsymbol{x}|y_i) = I(z_j; \boldsymbol{x}) - I(z_j; y_i) = 0$. On the other hand, we would like our metric for minimality to be in the range $[0, 1]$ for simplicity of interpretability and comparability with other metrics. Therefore, we define the minimality of $z_j$ with respect to $y_i$:

$$m_{ij} = 1 - \frac{I(z_j; \boldsymbol{x}|y_i)}{I(z_j; \boldsymbol{x})} = \frac{I(z_j; y_i)}{I(z_j; \boldsymbol{x})} \tag{5}$$

i.e., $m_{ij}$ collects the proportion of information about $z_j$ present in $y_i$ relative to that present in $\boldsymbol{x}$. Although it is useful to analyze $m_{ij}$ to obtain information about what is the connection between each factor and each variable, it is also interesting to have a single term that determines how minimal a representation $z$ is. We define this term below:

$$\bar{m} = \frac{1}{n_z} \sum_{j=1}^{n_z} \max_i m_{ij} \tag{6}$$

Thus, we are taking into account only the minimality for the factor that most influences each variable. Importantly, unlike other metrics in the literature, we do not compute the gap between $\max_i m_{ij}$ and other elements of $m_{ij}$, since this gap can be low for correlated factors even when $z_j$ is minimal: let $y_i$ and $y_{i'}$ two highly correlated factors, then $m_{ij} - m_{i'j} \approx 0$ even if $z_j$ is completely defined by $y_i$. From now on we refer to $\bar{m}$ as *Minimality* (italic) to differentiate it from minimality (lowercase and roman), which is the property defined in Proposition 1.

On the other hand, a representation $z_j$ is sufficient for $y_i$ when $I(y_i; \boldsymbol{x}|z_j) = 0$. Following a process analogous to the previous one, we define the sufficiency of $z_j$ with respect to $y_i$ as:

$$s_{ij} = 1 - \frac{I(y_i; \boldsymbol{x}|z_j)}{I(y_i; \boldsymbol{x})} = \frac{I(y_i; z_j)}{I(y_i; \boldsymbol{x})} \tag{7}$$

i.e., $s_{ij}$ captures the proportion of information about $y_j$ present in $z_j$ relative to that present in $\boldsymbol{x}$. Again, here we calculate a single value of sufficiency as:

$$\bar{s} = \frac{1}{n_y} \sum_{j=1}^{n_y} \max_j s_{ij} \tag{8}$$

From now on we refer to $\bar{s}$ as *Sufficiency* (italic) to differentiate it from sufficiency (lowercase and roman), which is the property defined in Proposition 2.

The problem with the above metrics is that it is in general intractable to compute $I(z_j; \boldsymbol{x})$ and $I(y_i; \boldsymbol{x})$, since this requires integrating over the entire space of entries. Similarly, it is difficult to find a good estimator of $I(z_j; \boldsymbol{x})$ and $I(y_i; \boldsymbol{x})$, since $\boldsymbol{x}$ is in general high-dimensional (Hausser & Strimmer, 2009). For this reason, we propose $\hat{m}_{ij}$ and $\hat{s}_{ij}$, which estimate $m_{ij}$ and $s_{ij}$ from a dataset, respectively. We provide these estimators in Algorithms 1 and 2 and derive them in Appendix B. Next, we give an intuition on why these estimators capture the minimality and sufficiency.

Given the set of inputs $\{\boldsymbol{x}^{(k)}\}$, the set of factors of variation $\{\boldsymbol{y}^{(k)}\} = \left\{\{y_i^{(k)}\}_{i=1}^n\right\}$, the set of representations $\{\boldsymbol{z}^{(k)}\} = \left\{\{z_j^{(k)}\}_{j=1}^m\right\}$ and the fact a representation $z_j$ is minimal with respect to $y_i$ if $I(z_j; \boldsymbol{x}|y_i) = 0$ or, equivalently, $p(z_j|\boldsymbol{x}) = p(z_j|y_i)$. Then, we can construct a regressor $f_{ij}$ that tries to predict $z_j^{(k)}$ from $y_i^{(k)}$ for each $k$. Subsequently, we compare the prediction $f_{ij}(y_i^{(k)}) \sim p(z_j|y_i^{(k)})$ with $z_j^{(k)} \sim p(z_j|\boldsymbol{x}^{(k)})$. When $y_i^{(k)}$ has no information about $z_j^{(k)}$, then $f_{ij}(y_i^{(k)})$ will tend to 0, which is the mean value of $z_j$ after standardization. Therefore, we will have that $\hat{m}_{ij} = 1 - var(z_j) = 0$. In the opposite case, when $y_i$ has all the information about $z_j$, we will have that $f_{ij}(y_i^{(k)}) = z_j^{(k)} \; \forall k$ and thus $\hat{m}_{ij} = 1$. To define $\hat{s}_{ij}$, we simply have to follow an analogous process taking into account that a representation $z_j$ is sufficient with respect to $y_i$ when $I(y_i; \boldsymbol{x}|z_j) = 0$. In all the experiments provided next, we have used random forest regressors for $f_{ij}$. We note that different families of regressors could be used and this could affect to the metrics.

| **Algorithm 1** Calculation of *Minimality* | **Algorithm 2** Calculation of *Sufficiency* |
|---|---|
| **Input:** $\left\{\{y_i^{(k)}\}_{i=1}^n\right\}, \left\{\{z_j^{(k)}\}_{j=1}^m\right\}$ | **Input:** $\left\{\{y_i^{(k)}\}_{i=1}^n\right\}, \left\{\{z_j^{(k)}\}_{j=1}^m\right\}$ |
| **Output:** $\bar{m}$ | **Output:** $\bar{s}$ |
| 1: $\{y_i^{(k)}\} \leftarrow \text{StandardScale}(\{y_i^{(k)}\}), i = 1$ to $n$ | 1: $\{y_i^{(k)}\} \leftarrow \text{StandardScale}(\{y_i^{(k)}\}), i = 1$ to $n$ |
| 2: $\{z_j^{(k)}\} \leftarrow \text{StandardScale}(\{z_j^{(k)}\}), j = 1$ to $m$ | 2: $\{z_j^{(k)}\} \leftarrow \text{StandardScale}(\{z_j^{(k)}\}), j = 1$ to $m$ |
| 3: **for** $j = 1$ to $m$ **do** | 3: **for** $i = 1$ to $n$ **do** |
| 4:     **for** $i = 1$ to $n$ **do** | 4:     **for** $j = 1$ to $m$ **do** |
| 5:         $f_{ij} \leftarrow \text{fit}\left(\{y_i^{(k)}\}, \{z_j^{(k)}\}\right)$ | 5:         $f_{ij} \leftarrow \text{fit}\left(\{z_j^{(k)}\}, \{y_i^{(k)}\}\right)$ |
| 6:         $\hat{m}_{ij} \leftarrow 1 - \frac{1}{K}\sum_k ||f_{ij}(y_i^{(k)}) - z_j^{(k)}||_2^2$ | 6:         $\hat{s}_{ij} \leftarrow 1 - \frac{1}{K}\sum_k ||f_{ij}(z_j^{(k)}) - y_i^{(k)}||_2^2$ |
| 7:     **end for** | 7:     **end for** |
| 8: **end for** | 8: **end for** |
| 9: $\bar{m} \leftarrow \frac{1}{m}\sum_{j=1}^n \max_i(\hat{m}_{ij})$ | 9: $\bar{s} \leftarrow \frac{1}{n}\sum_{i=1}^n \max_j(\hat{s}_{ij})$ |

## 5 EXPERIMENTS

We propose in this section a set of experiments in which we artificially define the representations based on the factors. This allows us to have exact knowledge of the relationship between the factors and the representation, so we can know with certainty if the different metrics are correctly capturing the presence or absence of disentanglement under different conditions.

### 5.1 DEFINING THE FACTORS OF VARIATION

In this experiment we define our own factors, nuisances and the relationship between these and the different variables of the representation. Thus, we can modify the degree of fulfillment of the properties of the section 3 and analyze whether different metrics capture these modifications.

**Minimality with Independent Factors of Variation** Here we analyze how the different metrics that are designed to measure minimality (or at least factors-invariance) behave in different scenarios. To do so, we design an experiment in which we have four factors $\boldsymbol{y} = (y_1, y_2, y_3, y_4)$ such that $y_i \sim \mathcal{U}[0, \pi)$ for $i = 1, 2, 3, 4$; a nuisance $n \sim \mathcal{U}[0, \pi)$, and a representation $\boldsymbol{z} = (z_1, z_2, z_3, z_4)$ such that $\boldsymbol{z} = \cos((1-\beta)A\boldsymbol{y} + \beta n)$, where $\beta \in [0, 1]$ and $A$ is defined as:

$$A = \begin{pmatrix} 1-\alpha & 0 & 0 & \alpha \\ \alpha & 1-\alpha & 0 & 0 \\ 0 & \alpha & 1-\alpha & 0 \\ 0 & 0 & \alpha & 1-\alpha \end{pmatrix} \tag{9}$$

where $\alpha \in [0, 0.5]$. Thus, the factors-invariance level will decrease as $\alpha$ grows and the nuisances-invariance level will decrease as $\beta$ grows. We should note that due to the injectivity of the cosine in $[0, \pi)$, each component of $\boldsymbol{z}$ can only have been generated by one element (or combination of these) of $\boldsymbol{y}$. In Figure 1 we compare different metrics for different values of $\alpha$ and $\beta$. First, we can see that $\beta$-VAE (Higgins et al., 2017), Factor-VAE (Kim et al., 2019), and Modularity (Ridgeway & Mozer, 2018) behaves approximately as the step-function for both $\alpha$ and $\beta$ so that they take high

values even when the levels of factors-invariance or nuisances-invariance are very low. Second, DCI-Disentanglement (Eastwood & Williams, 2018) only takes high values when factors-invariance and nuisances-invariance levels are very low. Thus, this metric takes low values even with high levels of minimality. Finally, *Minimality* (ours) palliates the above problems: it varies gradually according to the levels of factors-invariance and nuisances-invariance for a large range of values.

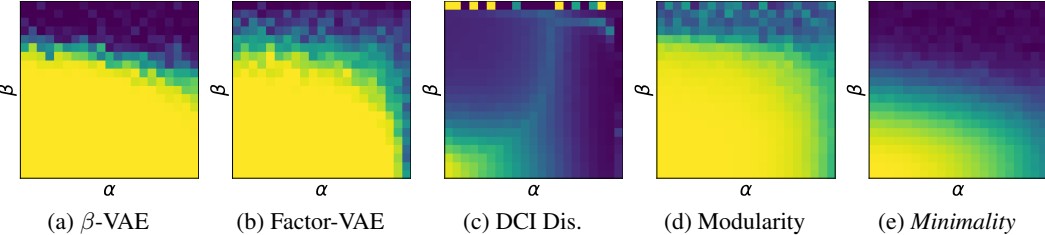

(a) $\beta$-VAE     (b) Factor-VAE     (c) DCI Dis.     (d) Modularity     (e) *Minimality*

Figure 1: Different metrics in the literature that aim to measure minimality for independent factors. In all the cases, $\alpha \in [0, 0.5]$ and $\beta \in [0, 1]$ and the values of the metrics go from 0 to 1.

**Sufficiency with Independent Factors of Variation**    Here we analyze how different metrics in the literature measure sufficiency (i.e. representations-invariance and explicitness) in an experiment similar to the previous one. In this case, we have four factors $\boldsymbol{y} = (y_1, y_2, y_3, y_4)$ such that $y_i \sim \mathcal{U}[0, \pi)$, and our representation $\boldsymbol{z} = \cos(\gamma A \boldsymbol{y})$, where $A$ is defined in equation 9 and $\gamma \in [1, 2)$. In this case, when $\alpha$ grows the representations-invariance level will decrease and when $\gamma$ grows, the explicitness level will decrease, because the cosine is not injective in $[0, \gamma\pi)$ for $\gamma > 1$ and there will be more than one possible combination of elements of $\boldsymbol{y}$ that can generate the components of $\boldsymbol{z}$ and we cannot completely obtain the factors from the $\boldsymbol{z}$ representation. In Figure 2 we compare our metric with others in the literature that try to measure sufficiency (or at least representations-invariance or explicitness) for different values of $\alpha$ and $\gamma$. First, we see that MIG (Chen et al., 2018) is practically insensitive to explicitness, since its value does not change with $\gamma$. Second, we see that SAP (Kumar et al., 2017) takes values close to 0 when $\alpha$ or $\gamma$ are equal to 0.5 and 2, respectively. However, these are not the minimum levels of representations-invariance and explicitness, so this metric is not too sensitive for low values of sufficiency. Third, we see that for a high range of values of $\alpha$, DCI-Completeness (Eastwood & Williams, 2018) takes higher values the higher $\gamma$ is, i.e. its value increases the less explicit the representation is. Fourth, we see that when $\alpha$ is low, Explicitness (Ridgeway & Mozer, 2018) takes high values even when $\gamma$ is high, i.e. it gives low values of explicitness when they are actually high. Finally, we see that *Sufficiency* (ours) mitigates all the problems mentioned in the previous metrics varying gradually with the level of representations-invariance and explicitness.

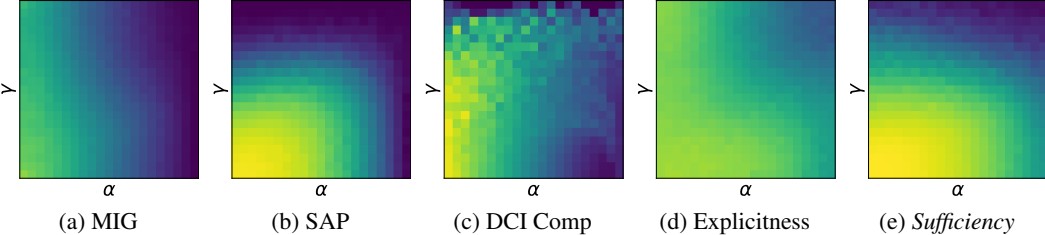

(a) MIG     (b) SAP     (c) DCI Comp     (d) Explicitness     (e) *Sufficiency*

Figure 2: Different metrics in the literature that aim to measure sufficiency for independent factors. In all the cases, $\alpha \in [0, 0.5]$ and $\gamma \in [1, 2)$ and the values of the metrics go from 0 to 1.

**Minimality with Dependent Factors of Variation**    To compare the behavior of the different metrics in the condition of non-independent factors, we design an experiment similar to the previous one, but with slight modifications. On the one hand, we define $y_1$ and $y_2$ in the same way as in the previous case, but $y_3 = (1 - \delta)y_1 + \delta y_2$ and $y_4 = (1 - \delta)y_2 + \delta y_1$, where $\delta \in [0, 0.5]$. On the other hand, we define $\boldsymbol{z} = \cos(A\boldsymbol{y})$, where $A$ is the matrix in equation 9. In this case, we do not include nuisances, since having nuisances is independent of whether or not the factors are correlated. In Figure 3 we show the values of different metrics according to $\alpha$ and $\delta$. In this case, we see that when $\alpha = 0$, the only metric that gives a maximum value for all $\delta$ is *Minimality* (ours). That is, when $A$ is the identity matrix and we have one-to-one ratio of each variable with a factor, the rest of the metrics can give values different from one. In particular, we see that $\beta$-VAE (Higgins et al., 2017) has again a step-like behavior, Factor-VAE (Kim et al., 2019) is quite sensitive to the correlation

strength of the factors, and DCI-Disentanglement (Eastwood & Williams, 2018) and Modularity (Ridgeway & Mozer, 2018) give low and intermediate values for almost any combination of values. We should note that *Minimality* (ours) takes high values in a large part of the cases since the level of factors-invariance is in general high and the level of nuisances-invariance is always maximum.

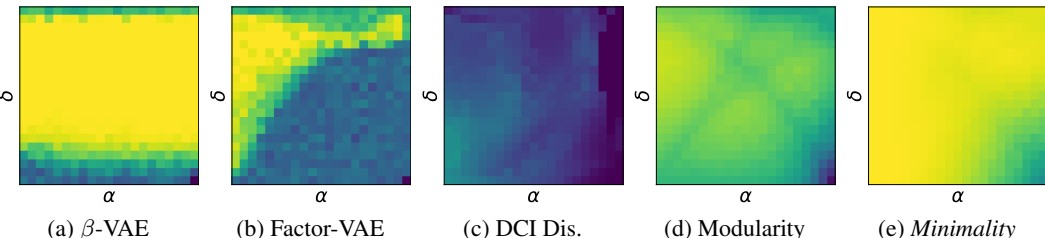

(a) $\beta$-VAE      (b) Factor-VAE      (c) DCI Dis.      (d) Modularity      (e) *Minimality*

Figure 3: Different metrics in the literature that aim to measure minimality for dependent factors. In all the cases, $\alpha \in [0, 0.5]$ and $\delta \in [0, 0.5]$ and the values of the metrics go from 0 to 1.

**Sufficiency with Dependent Factors of Variation**    This experiment is identical to the one described in the previous paragraph and the goal is to analyze how the different metrics that try to measure sufficiency behave depending on the level of the correlation between the factors. The results are shown in Figure 4. We see that the values of MIG (Chen et al., 2018) and SAP (Kumar et al., 2017) are low regardless of $\alpha$ and $\delta$. On the other hand, DCI-Completeness (Eastwood & Williams, 2018) does not take a maximum value whenever $\alpha = 0$ even though in this case it is possible to predict all factors from a single variable. Finally, Explicitness (Ridgeway & Mozer, 2018) does not take a maximum value even when the factors can be completely defined by the representation. Finally, we have that *Sufficiency* (ours) takes maximum values when the representations-invariance and explicitness are maximal and gradually decreases with these.

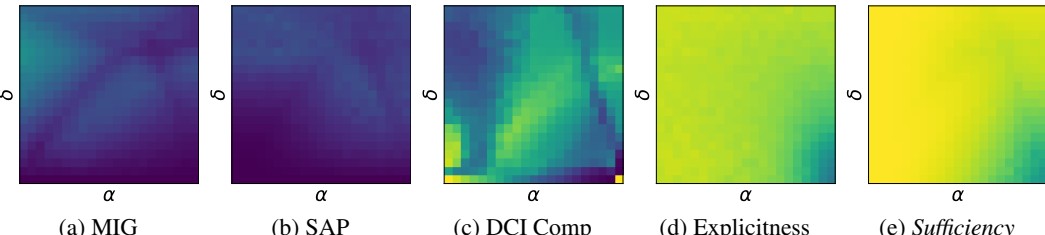

(a) MIG      (b) SAP      (c) DCI Comp      (d) Explicitness      (e) *Sufficiency*

Figure 4: Different metrics in the literature that aim to measure sufficiency for dependent factors. In all the cases, $\alpha \in [0, 0.5]$ and $\delta \in [0, 0.5]$ and the values of the metrics go from 0 to 1.

## 5.2   LITERATURE DATASETS

As mentioned, the factors in most of the datasets in the literature for measuring disentanglement are independent. However, Träuble et al. (2021) propose a modification to introduce correlation between pairs of factors as $p(y_1, y_2) \propto \exp\left(-(y_1 - \alpha y_2)^2 / 2\sigma^2\right)$, where $\alpha = c_2^{\max} / c_2^{\max}$, i.e., the lower the value of $\sigma$ the stronger the correlation. Subsequently, Roth et al. (2022) propose to introduce this correlation scheme between multiple pairs of factors. Here, we propose a similar scheme. Concretely, the results include scenarios with uncorrelated factors (original datasets), correlations between multiple pairs of factors (a factor is never correlated with more than one factor) and shared confounders (one factor correlates to all others). As in the previous experiment, we artificially define the representation from the factors, so that we can have exact knowledge of the level of disentanglement. We define $\mathbf{z} = A\mathbf{y}$, so that $A = (a_{ij})$, $a_{ii} = 1 - \alpha$, $a_{ij} \sim \mathcal{U}[0, \alpha]$ if $i \neq j$ and $\alpha \in [0, 0.5]$. Thus, the representation will be fully disentangled when $\alpha = 0$. We use this scheme for Shapes3D (Burgess & Kim, 2018), dSprites (Matthey et al., 2017), MPI3D (Gondal et al., 2019) and CelebA(Liu et al., 2015). The results for MPI3D are shown next and those corresponding to Shapes3D, dSprites and CelebA can be found in Appendix D due to space limitations and the fact that the conclusions drawn are similar to those of MPI3D. We compute the experiments 10 times and values of mean and standard deviation are provided in the figures. More details about the experiment (concretely, values of $\sigma$ and correlated pairs) can also be found in Appendix D.

In Figure 5 we see the values of the different metrics that try to measure minimality for the different levels of correlation. First and most importantly, we see that *Minimality* (ours) is the only metric that is always equal to 1 when $\alpha = 0$. This means that the other metrics fail in detecting perfect disentanglement even in the simplest case (i.e., $z = y$) under correlated factors of variation. Second, we find similar values of *Minimality* for 1, 2 and 3 pairs of correlated factors and the different values of $\alpha$. This is desirable, since $\alpha$ is what determines the degree of minimality irrespective of the number of pairs of factors. Finally, in the case of a shared confounder, our metric is the only one taking its near-maximum value regardless of the value of $\alpha$. Again, this is desirable, since one factor is strongly confounding to all the others, which translates into the fact that each variable is affected by only "almost" one factor.

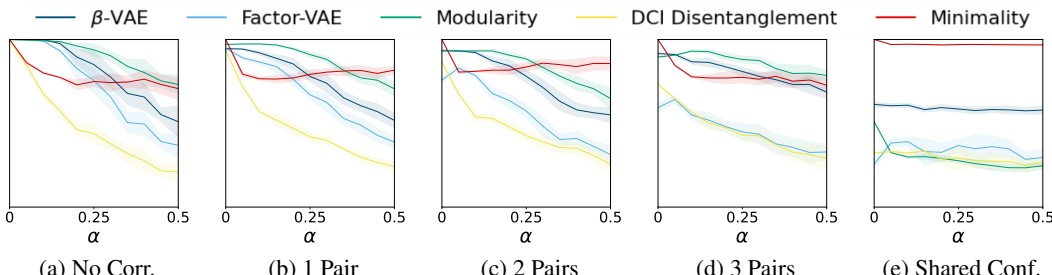

(a) No Corr.      (b) 1 Pair      (c) 2 Pairs      (d) 3 Pairs      (e) Shared Conf.

Figure 5: Metrics measuring minimality for MPI3D. The values of all the metrics go from 0 to 1.

In Figure 6 we analyze the different metrics to measure sufficiency in the previous scenarios. First, we can see that *Sufficiency* (ours) is the only metric that is equal to 1 always that $\alpha = 0$. Again, the other metrics fail in detecting maximum disentanglement under the presence of correlated factors of variation. Second, the values of *Sufficiency* decrease slower with $\alpha$ when the level of correlation between factors is higher. This is desirable because the fact that factors are more correlated implies that, for the same value of $\alpha$, a variable $z_j$ will tend to have more information about any factor $y_k$ through its "corresponding" factor $y_i$. The extreme of this is in the case of a shared confounder, in which *Sufficiency* takes almost its maximum value regardless of $\alpha$, since we can describe every factor "almost" perfectly by using only one variable due to the high correlation between factors.

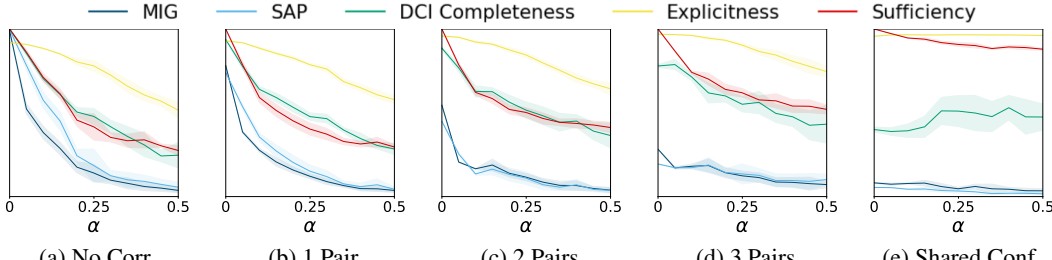

(a) No Corr.      (b) 1 Pair      (c) 2 Pairs      (d) 3 Pairs      (e) Shared Conf.

Figure 6: Metrics measuring sufficiency for MPI3D. The values of all the metrics go from 0 to 1.

## 6 CONCLUSION

In this work we have presented two problems that the different definitions of disentanglement suffer: (i) they do not analyze the invariance to the nuisances; and (ii) they assume that the factors of variation are independent of each other and of the nuisances. Consequently, the metrics used to measure disentanglement do not capture it accurately when any of these two factors are present. To address these problems, we have first proposed four properties from the point of view of information theory that serve to define a disentangled representation considering the nuisances in the input and the scenario in which the factors can be dependent. Furthermore, we have related these properties to the concepts of minimality and sufficiency of a representation. In fact, we have argued that it is more convenient to measure the degree of sufficiency and minimality of a representation rather than measuring the four properties individually. Subsequently, we have proposed metrics to measure sufficiency and minimality and derived estimators of them. Finally, we have compared our metrics with others in the literature to illustrate that our metrics are able to correctly capture the level of disentanglement under a wider variety of scenarios.

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

## A    PROOF OF PROPOSITIONS

**Proposition 1.** *Let $\boldsymbol{y} = \{y_l\}_{l=1}^n$ be some factors and $\boldsymbol{z} = \{z_j\}_{j=1}^m$ a representation. If $z_j$ is a minimal representation of $y_i$, then $z_j$ is factors-invariant for $y_i$ and nuisances-invariant. Equivalently, we have that:*

$$(\boldsymbol{x} \leftrightarrow y_i \leftrightarrow z_j) \iff (\tilde{\boldsymbol{y}}_i \leftrightarrow y_i \leftrightarrow z_j) \wedge (\boldsymbol{n} \leftrightarrow y_i \leftrightarrow z_j)$$

*Proof.* First, we demonstrate that $(\boldsymbol{x} \leftrightarrow y_i \leftrightarrow z_j) \implies (\tilde{\boldsymbol{y}}_i \leftrightarrow y_i \leftrightarrow z_j)$: Given this Markov chain, by the DPI, we have that $I(z_j; \boldsymbol{x}) \leq I(z_j; y_i)$. Since $z_j$ is a representation of $\boldsymbol{x}$, we also have by DPI that $I(z_j; \boldsymbol{y}) \leq I(z_j; \boldsymbol{x})$. Therefore, it follows that $I(z_j; \boldsymbol{y}) \leq I(z_j; y_i)$ On the other hand, by the chain rule of mutual information, we have that $I(z_j; \boldsymbol{y}) = I(z_j; y_i, \tilde{\boldsymbol{y}}_i) = I(z_j; y_i) + I(z_j; \tilde{\boldsymbol{y}}_i|y_i)$. By the non-negativity of mutual information, we are left with $I(z_j; \boldsymbol{y}) \geq I(z_j; y_i)$. Therefore, we have that $I(z_j; y_i) = I(z_j; \boldsymbol{y}) = I(z_j; y_i, \tilde{\boldsymbol{y}}_i)$ or, equivalently, $\tilde{\boldsymbol{y}}_i \leftrightarrow y_i \leftrightarrow z_j$.

Second, we demonstrate that $(\boldsymbol{x} \leftrightarrow y_i \leftrightarrow z_j) \implies (\boldsymbol{n} \leftrightarrow y_i \leftrightarrow z_j)$: Given this Markov chain, we know from the DPI that $I(z_j; \boldsymbol{x}) \leq I(z_j; y_i)$. On the other hand, since $z_j$ is a representation of $\boldsymbol{x}$, we have the Markov chain $(\boldsymbol{n}, y_i) \leftrightarrow \boldsymbol{x} \leftrightarrow z_j$. Equivalently, we have that $I(z_j; \boldsymbol{n}, y_i) \leq I(z_j; \boldsymbol{x})$. The chain rule of mutual information tells us that $I(z_j; \boldsymbol{n}|y_i) = I(z_j; \boldsymbol{n}, y_i) - I(z_j|y_i)$. According to the above two points, we are left with $I(z_j; \boldsymbol{n}|y_i) \leq I(z_j; \boldsymbol{x}) - I(z_j|y_i)$. Therefore, because of the non-negativity of mutual information, it is only possible that $I(z_j; \boldsymbol{n}|y_i) = 0$ or, equivalently, $\boldsymbol{n} \leftrightarrow y_i \leftrightarrow z_j$.

Finally, we demonstrate that $(\tilde{\boldsymbol{y}}_i \leftrightarrow y_i \leftrightarrow z_j) \wedge (\boldsymbol{n} \leftrightarrow y_i \leftrightarrow z_j) \implies (\boldsymbol{x} \leftrightarrow y_i \leftrightarrow z_j)$: We know by the Bayes' theorem that $p(z_j, \boldsymbol{y}, \boldsymbol{n}) = p(z_j|\boldsymbol{y}, \boldsymbol{n})p(\boldsymbol{n}|\boldsymbol{y})p(\boldsymbol{y}) = p(\boldsymbol{n}|z_j, \boldsymbol{y})p(z_j|\boldsymbol{y})p(\boldsymbol{y}) = p(\boldsymbol{n}|z_j, \boldsymbol{y})p(z_j|y_i)p(\boldsymbol{y})$. Thus, $p(z_j|x) = p(z_j|\boldsymbol{y}, \boldsymbol{n}) = p(z_j|y_i)$ if and only if $p(\boldsymbol{n}|\boldsymbol{y}) = p(\boldsymbol{n}|z_j, \boldsymbol{y})$, which is equivalent to having $p(z_j|\boldsymbol{y}) = p(z_j|\boldsymbol{y}, \boldsymbol{n}) = p(z_j|\boldsymbol{x})$ and since $p(z_j|\boldsymbol{y}) = p(z_j|y_i)$, we are left with $p(z_j|\boldsymbol{x}) = p(z_j|y_i)$, and thus, $(\boldsymbol{x} \leftrightarrow y_i \leftrightarrow z_j)$. $\square$

**Proposition 2.** *Let $y_i$ be a factor and $\boldsymbol{z} = \{z_l\}_{l=1}^m$ a representation. Then, $z_j$ is a sufficient representation of $y_i$ if and only if $z_j$ is representations-invariant for $y_i$ and $\boldsymbol{z}$ is explicit for $y_i$. Equivalently, we have the that:*

$$(\boldsymbol{x} \leftrightarrow z_j \leftrightarrow y_i) \iff (\tilde{\boldsymbol{z}}_j \leftrightarrow z_j \leftrightarrow y_i) \wedge (\boldsymbol{x} \leftrightarrow \boldsymbol{z} \leftrightarrow y_i)$$

*Proof.* First, we demonstrate that $(\boldsymbol{x} \leftrightarrow z_j \leftrightarrow y_i) \implies (\tilde{\boldsymbol{z}}_j \leftrightarrow z_j \leftrightarrow y_i)$: Given this Markov chain, by the DPI, we have that $I(y_i; \boldsymbol{x}) \leq I(y_i; z_j)$. Since $\boldsymbol{z}$ is a representation of $\boldsymbol{x}$, we also have that $I(y_i; \boldsymbol{z}) \leq I(y_i; \boldsymbol{x})$. Therefore, it follows that $I(y_i; \boldsymbol{z}) \leq I(y_i; z_j)$ On the other hand, by the mutual information chain rule, we have that $I(y_i; \boldsymbol{z}) = I(y_i; z_j, \tilde{\boldsymbol{z}}_j) = I(y_i; z_j) + I(y_i; \tilde{\boldsymbol{z}}_j|z_j)$. By the non-negativity of mutual information, we are left with $I(y_i; \boldsymbol{z}) \geq I(y_i; z_j)$. Therefore, we have that $I(y_i; z_j) = I(y_i; \boldsymbol{z}) = I(y_i; z_j, \tilde{\boldsymbol{z}}_j)$ or, equivalently, $\tilde{\boldsymbol{z}}_j \leftrightarrow z_j \leftrightarrow y_i$.

Second, we demonstrate that $(\boldsymbol{x} \leftrightarrow z_j \leftrightarrow y_i) \implies (\boldsymbol{x} \leftrightarrow \boldsymbol{z} \leftrightarrow y_i)$: Since $z_j$ is a representation of $\boldsymbol{x}$, we have that $p(y_i|\boldsymbol{x}, z_j) = p(y_i|\boldsymbol{x})$. Moreover, since $z_j$ is sufficient, we have that $p(y_i|\boldsymbol{x}, z_j) = p(y_i|z_j)$. Putting the above two terms together, we are left with $p(y_i|\boldsymbol{x}) = p(y_i|z_j)$. As we have already shown, if $z_j$ is sufficient, then we have the Markov chain $(\tilde{\boldsymbol{z}}_j \leftrightarrow z_j \leftrightarrow y_i)$, so $p(y_i|\boldsymbol{z}) = p(y_i|z_j)$. Therefore, we are left with $p(y_i|\boldsymbol{z}) = p(y_i|\boldsymbol{x})$. Finally, since $\boldsymbol{z}$ is a representation of $\boldsymbol{x}$, we know that $p(y_i|\boldsymbol{x}) = p(y_i|\boldsymbol{x}, \boldsymbol{z})$. Putting all of the above together, we are left with $p(y_i|\boldsymbol{x}, \boldsymbol{z}) = p(y_i|\boldsymbol{z})$ or, equivalently, $\boldsymbol{x} \leftrightarrow \boldsymbol{z} \leftrightarrow y_i$.

Finally, we demonstrate that $(\tilde{\boldsymbol{z}}_j \leftrightarrow z_j \leftrightarrow y_i) \wedge (\boldsymbol{x} \leftrightarrow \boldsymbol{z} \leftrightarrow y_i) \implies (\boldsymbol{x} \leftrightarrow z_j \leftrightarrow y_i)$: Since $z_j$ and $\boldsymbol{z}$ are representations of $\boldsymbol{x}$, we have that $p(y_i|\boldsymbol{x}, z_j) = p(y_i|\boldsymbol{x}, \boldsymbol{z}) = p(y_i|\boldsymbol{x})$. Furthermore, given the Markov chain $\boldsymbol{x} \leftrightarrow \boldsymbol{z} \leftrightarrow y_i$, we know that $p(y_i|\boldsymbol{x}, \boldsymbol{z}) = p(y_i|\boldsymbol{z})$. Equivalently, given $\tilde{\boldsymbol{z}}_j \leftrightarrow z_j \leftrightarrow y_i$, we have that $p(y_i|z_j) = p(y_i|z_j, \tilde{\boldsymbol{z}}_j) = p(y_i|\boldsymbol{z})$. Recapitulating the above, we are left with $p(y_i|z_j) = p(y_i|\boldsymbol{z}) = p(y_i|\boldsymbol{x}, \boldsymbol{z}) = p(y_i|\boldsymbol{x}, z_j)$. Therefore, we have that $\boldsymbol{x} \leftrightarrow z_j \leftrightarrow y_i$. $\square$

# B    DERIVATION OF THE ESTIMATORS FOR *Minimality* AND *Sufficiency*

In section 4.3 we have defined metrics to measure the levels of minimality and sufficiency, but their exact calculation is in general computationally intractable. Moreover, they include mutual information terms of the input $\boldsymbol{x}$ with other variables, so there are no good estimators either, since $\boldsymbol{x}$ is in general high dimensional. We derive next the estimators described in Algorithms 1 and 2.

## B.1    DERIVATION OF THE ESTIMATOR FOR *Minimality*

As explained, we define the minimality of a variable $z_j$ with respect to a factor $y_i$ as:

$$m_{ij} = 1 - \frac{I(z_j; \boldsymbol{x}|y_i)}{I(z_j; \boldsymbol{x})} = \frac{I(z_j; y_i)}{I(z_j; \boldsymbol{x})} \tag{10}$$

i.e., $m_{ij}$ collects the proportion of information about $z_j$ present in $y_i$ relative to that present in $\boldsymbol{x}$. The problem with this metric is that it is in general intractable to compute $I(z_j; \boldsymbol{x})$ and it is difficult to find a good estimator of $I(z_j; \boldsymbol{x})$, since $\boldsymbol{x}$ is in general high-dimensional.

In the following, we derive approximations for $I(z_j; \boldsymbol{x}|y_i)$.

$$I(z_j; \boldsymbol{x}|y_i) = \iiint p(\boldsymbol{x}, y_i, z_j) \log \frac{p(\boldsymbol{x}, y_i, z_j) p(y_i)}{p(\boldsymbol{x}, y_i) p(y_i, z_j)} \, d\boldsymbol{x} \, dy_i \, dz_j \tag{11}$$

$$= \iiint p(\boldsymbol{x}, y_i) p(z_j | \boldsymbol{x}, y_i) \log \frac{p(z_j | \boldsymbol{x}, y_i)}{p(z_j | y_i)} \, d\boldsymbol{x} \, dy_i \, dz_j \tag{12}$$

$$= \iiint p(\boldsymbol{x}, y_i) p(z_j | \boldsymbol{x}) \log \frac{p(z_j | \boldsymbol{x})}{p(z_j | y_i)} \, d\boldsymbol{x} \, dy_i \, dz_j \tag{13}$$

$$\approx \frac{1}{|D|} \sum_{\left(\boldsymbol{x}^{(k)}, y_i^{(k)}\right) \in \mathcal{D}} \int p(z_j | \boldsymbol{x}^{(k)}) \log \frac{p\left(z_j | \boldsymbol{x}^{(k)}\right)}{p\left(z_j | y_i^{(k)}\right)} \, dz_j \tag{14}$$

$$= \frac{1}{|D|} \sum_{\left(\boldsymbol{x}^{(k)}, y_i^{(k)}\right) \in \mathcal{D}} D_{KL}\left(p\left(z_j | \boldsymbol{x}^{(k)}\right) \,\Big\|\, p\left(z_j | y_i^{(k)}\right)\right) = \hat{I}(z_j; \boldsymbol{x}|y_i) \tag{15}$$

On the other hand, as described in different papers (Bishop, 1994), the output of a regression or classification system (a linear model or a neural network, for example) are not necessarily predictions of the value of the variable output but the parameters of the distribution of the variable output. Therefore, we may assume without loss of generality that $p\left(z_j | \boldsymbol{x}^{(k)}\right) = \mathcal{N}\left(z_j; f_j\left(\boldsymbol{x}^{(k)}, \theta\right), I\right)$, where $f$ is our representation learning system and $\theta$ its parameters. That is, our representation learning system is not predicting a representation for each input but the mean of the distribution of $z_j$ for a given input $\boldsymbol{x}^{(k)}$. Therefore, we would not have one representation per input but infinite. Equivalently, we can assume that $p\left(z_j | y_i^{(k)}\right) = \mathcal{N}\left(z_j; f_{ij}\left(y_i^{(k)}, \phi\right), I\right)$, where $f_{ij}$ is a regressor whose parameters are $\phi$ that predicts the mean of $z_j$ factor $y_i^{(k)}$. Under this assumption, we have that:

$$D_{KL}\left(p\left(z_j | \boldsymbol{x}^{(k)}\right) \,\Big\|\, p\left(z_j | y_i^{(k)}\right)\right) = \frac{1}{2} \left\| f_j\left(\boldsymbol{x}^{(k)}, \theta\right) - f_{ij}\left(y_i^{(k)}, \phi\right) \right\|_2^2 \tag{16}$$

Thus, we have that

$$\hat{I}(z_j; \boldsymbol{x}|y_i) = \frac{1}{2|D|} \sum_{\left(\boldsymbol{x}^{(k)}, y_i^{(k)}\right) \in \mathcal{D}} \left\| f_j\left(\boldsymbol{x}^{(k)}, \theta\right) - f_{ij}\left(y_i^{(k)}, \phi\right) \right\|_2^2 \tag{17}$$

Through a procedure analogous to the previous one, we give an approximation for the term $I(z_j; x)$:

$$I(z_j; \boldsymbol{x}) = \iint p(\boldsymbol{x}, z_j) \log \frac{p(\boldsymbol{x}, z_j)}{p(\boldsymbol{x})p(z_j)} \, d\boldsymbol{x} \, dz_j \tag{18}$$

$$= \iint p(\boldsymbol{x})p(z_j|\boldsymbol{x}) \log \frac{p(z_j|x)}{p(z_j)} \, d\boldsymbol{x} \, dz_j \tag{19}$$

$$\approx \frac{1}{|D|} \sum_{\boldsymbol{x}^{(k)} \in \mathcal{D}} D_{KL}\left( p\left(z_j|\boldsymbol{x}^{(k)}\right) \,\middle\|\, p\left(z_j\right) \right) = \tilde{I}(z_j; \boldsymbol{x}) \tag{20}$$

On the other hand, we have that:

$$p(z_j) = \int p(\boldsymbol{x}, z_j) \, dx \approx \frac{1}{|D|} \sum_{x^{(l)} \in \mathcal{D}} p\left(z_j|\boldsymbol{x}^{(l)}\right) \tag{21}$$

that is, $p(z_j)$ can be approximated as a mixture of $|\mathcal{D}|$ Gaussians with the identity matrix as its covariance matrix. Since $D_{KL}(p\|q)$ is a convex function on the pair $(p, q)$, we have that:

$$\tilde{I}(z_j; \boldsymbol{x}) \approx \frac{1}{|D|} \sum_{\boldsymbol{x}^{(k)} \in \mathcal{D}} D_{KL}\left( p\left(z_j|\boldsymbol{x}^{(k)}\right) \,\middle\|\, \frac{1}{|D|} \sum_{x^{(l)} \in \mathcal{D}} p\left(z_j|\boldsymbol{x}^{(l)}\right) \right) \tag{22}$$

$$\approx \frac{1}{|D|} \sum_{\boldsymbol{x}^{(k)} \in \mathcal{D}} \frac{1}{|D|} \sum_{\boldsymbol{x}^{(l)} \in \mathcal{D}} D_{KL}\left( p\left(z_j|\boldsymbol{x}^{(k)}\right) \,\middle\|\, p\left(z_j|\boldsymbol{x}^{(l)}\right) \right) \tag{23}$$

$$= \frac{1}{2|D|^2} \sum_{\boldsymbol{x}^{(k)} \in \mathcal{D}} \sum_{\boldsymbol{x}^{(l)} \in \mathcal{D}} \left\| f_j\left(\boldsymbol{x}^{(k)}, \theta\right) - f_j\left(\boldsymbol{x}^{(l)}, \theta\right) \right\|_2^2 \tag{24}$$

$$= \frac{1}{2|D|} \sum_{\boldsymbol{x}^{(k)} \in \mathcal{D}} \left\| f_j\left(\boldsymbol{x}^{(k)}, \theta\right) \right\|_2^2 - \frac{1}{2|D|^2} \sum_{\boldsymbol{x}^{(k)} \in \mathcal{D}} f_j\left(\boldsymbol{x}^{(k)}, \theta\right)^T \sum_{\boldsymbol{x}^{(l)} \in \mathcal{D}} f_j\left(\boldsymbol{x}^{(l)}, \theta\right) \tag{25}$$

$$= \frac{1}{2|D|} \sum_{\boldsymbol{x}^{(k)} \in \mathcal{D}} \left\| f_j\left(\boldsymbol{x}^{(k)}, \theta\right) \right\|_2^2 - \left\| \bar{f}_j\left(\boldsymbol{x}, \theta\right) \right\|_2^2 = \hat{I}(z_j; \boldsymbol{x}) \tag{26}$$

where line 23 is demonstrated in Proposition 3 and $\bar{f}_j(\boldsymbol{x}, \theta)$ is the sample mean of $f_j(\boldsymbol{x}, \theta)$. We must note that if we standardize the samples from $f_j(\boldsymbol{x}, \theta)$ to have zero mean and unit variance, then we have that:

$$\hat{I}(z_j; \boldsymbol{x}) = \frac{1}{|D|} \sum_{\boldsymbol{x}^{(k)} \in \mathcal{D}} \left\| f_j\left(\boldsymbol{x}^{(k)}, \theta\right) \right\|_2^2 = 1 \tag{27}$$

Finally, we define $\tilde{m}_{ij}$, which is an estimator of $m_{ij}$:

$$\hat{m}_{ij} = 1 - \frac{\hat{I}(z_j; \boldsymbol{x}|y_i)}{\hat{I}(z_j; \boldsymbol{x})} \tag{28}$$

$$= 1 - \frac{1}{|D|} \sum_{\left(x^{(k)}, y_i^{(k)}\right) \in \mathcal{D}} \left\| f_j\left(x^{(k)}, \theta\right) - f_{ij}\left(y_i^{(k)}, \phi\right) \right\|_2^2 \tag{29}$$

**Proposition 3.** *Let $Q = \{q_i(z)\}_{i=1}^{n}$ be a set of distributions such that $q_i(z) = \mathcal{N}(z; \mu_i, I)$ and the set $\{\mu_i\}_{i=1}^{n}$ is uniformly distributed with zero mean and unit variance. Thus we have that:*

$$\sum_i D_{KL}\left(q_i(z) \,\middle\|\, \frac{1}{n}\sum_k q_k(z)\right) \approx \sum_i \frac{1}{n}\sum_k D_{KL}\left(q_i(z) \,\middle\|\, q_k(z)\right)$$

*Proof.* We have that:

$$\sum_i D_{KL}\left(q_i(z) \,\middle\|\, \frac{1}{n}\sum_k q_k(z)\right) = \sum_i \int q_i(z)\log\frac{q_i(z)}{\frac{1}{n}\sum_k q_k(z)}\,dz \tag{30}$$

$$= \sum_i \int \frac{1}{n}\sum_j q_i(z)\log\left(\frac{q_i(z)}{q_j(z)}\frac{q_j(z)}{\frac{1}{n}\sum_k q_k(z)}\right)dz \tag{31}$$

$$= \sum_i \frac{1}{n}\sum_j D_{KL}\left(q_i(z)\,\middle\|\,q_j(z)\right) \tag{32}$$

$$+ \frac{1}{n}\sum_i\sum_j \int q_i(z)\log\frac{nq_j(z)}{\sum_k q_k(z)}\,dz \tag{33}$$

Thus we have that the the proposition is true when:

$$\frac{1}{n}\sum_i\sum_j \int q_i(z)\log\frac{nq_j(z)}{\sum_k q_k(z)}\,dz \approx 0 \tag{34}$$

We can reformulate this term as:

$$\frac{1}{n}\mathbb{E}_{z\in q_i(z)}\left[\mathbb{E}_{q_i\in Q}\left[\sum_j \log\frac{nq_j(z)}{\sum_k q_k(z)}\right]\right] \tag{35}$$

Due to the Central Limit Theorem, we have this is almost equal to:

$$\frac{1}{n}\mathbb{E}_{z\in\mathcal{N}(0,I)}\left[\sum_j \log\frac{nq_j(z)}{\sum_k q_k(z)}\right] \tag{36}$$

If we simply sample once so that $z = 0$, we have that we can approximate the previous term as:

$$\frac{1}{n}\sum_j \log\frac{n\exp\left(-\frac{1}{2}||\mu_j||_2^2\right)}{\sum_k \exp\left(-\frac{1}{2}||\mu_k||_2^2\right)} \tag{37}$$

Since, $\{\mu_i\}_{i=1}^{n}$ is uniformly distributed, the set of their norms is also uniformly distributed and this term is zero. $\qquad\square$

### B.2 DERIVATION OF THE ESTIMATOR FOR *Sufficiency*

As explained, we define the sufficiency of a variable $z_j$ for a factor $y_i$ as:

$$s_{ij} = 1 - \frac{I(y_i; \boldsymbol{x}|z_j)}{I(y_i; \boldsymbol{x})} = \frac{I(y_i; z_j)}{I(y_i; \boldsymbol{x})} \tag{38}$$

i.e., $s_{ij}$ captures the proportion of information about $y_j$ present in $z_j$ relative to that present in $\boldsymbol{x}$. We have the same problem as in the previous case: it is in general intractable to compute $I(y_i; \boldsymbol{x})$ and it is difficult to find a good estimator of $I(y_i; \boldsymbol{x})$, since $\boldsymbol{x}$ is in general high-dimensional.

Following a process analogous to that of the minimality case, we arrive at the estimator:

$$\hat{I}(y_i; \boldsymbol{x}|z_j) = \frac{1}{|D|}\sum_{\left(\boldsymbol{x}^{(k)}, y_i^{(k)}\right)\in\mathcal{D}} D_{KL}\left(p\left(y_i|\boldsymbol{x}^{(k)}\right)\,\middle\|\,p\left(y_i|z_j^{(k)}\right)\right) \tag{39}$$

Again, we can assume that the label $l_i^{(k)}$ of the factor $y_i^{(k)}$ is its mean and that $p\left(y_i|\boldsymbol{x}^{(k)}\right) = \mathcal{N}\left(y_i; l_i, I\right)$. As in the minimality estimator, we can assume that $p\left(y_i|z_j^{(k)}\right) = \mathcal{N}\left(y_i; f_{ij}\left(z_j^{(k)}, \phi\right), I\right)$, where $f_{ij}$ is a regressor whose parameters are $\phi$ that predicts the mean of $y_i$ from the variable $z_j^{(k)}$. Under this assumption, we have that:

$$D_{KL}\left(p\left(y_i|\boldsymbol{x}^{(k)}\right) \middle\| p\left(y_i|z_j^{(k)}\right)\right) = \frac{1}{2}\left\|l_i^{(k)} - f_{ij}\left(z_j^{(k)}, \phi\right)\right\|_2^2 \tag{40}$$

Through a procedure analogous to the previous one, we give an approximation for the term $I(z_j; \boldsymbol{x})$:

$$I(y_i; \boldsymbol{x}) = \iint p(\boldsymbol{x}, y_i) \log \frac{p(\boldsymbol{x}, y_i)}{p(\boldsymbol{x})p(y_i)} \, d\boldsymbol{x} \, dy_i \tag{41}$$

$$\approx \frac{1}{|D|} \sum_{\boldsymbol{x}^{(k)} \in \mathcal{D}} D_{KL}\left(p\left(y_i|\boldsymbol{x}^{(k)}\right) \middle\| p\left(y_i\right)\right) = \tilde{I}(y_i; \boldsymbol{x}) \tag{42}$$

And we can approximate as:

$$p(y_i) = \int p(\boldsymbol{x}, y_i) \, d\boldsymbol{x} \approx \frac{1}{|D|} \sum_{\boldsymbol{x}^{(l)} \in \mathcal{D}} p\left(y_i|\boldsymbol{x}^{(l)}\right) \tag{43}$$

Again, we have that:

$$\tilde{I}(y_i; \boldsymbol{x}) \approx \frac{1}{|D|} \sum_{\boldsymbol{x}^{(k)} \in \mathcal{D}} \frac{1}{|D|} \sum_{\boldsymbol{x}^{(l)} \in \mathcal{D}} D_{KL}\left(p\left(y_i|\boldsymbol{x}^{(k)}\right) \middle\| p\left(y_i|\boldsymbol{x}^{(l)}\right)\right) \tag{44}$$

$$= \frac{1}{2|D|} \sum_{l_i^{(k)} \in \mathcal{D}} \left\|l_i^{(k)}\right\|_2^2 - \left\|\bar{l}_i^{(k)}\right\|_2^2 = \hat{I}(z_j; \boldsymbol{x}) \tag{45}$$

where $\bar{l}_i$ is the sample mean of $l_i$. We must note that if we standardize the samples from $l_i$ to have zero mean and unit variance, then we have that:

$$\hat{I}(y_i; \boldsymbol{x}) = \frac{1}{|D|} \sum_{l_i^{(k)} \in \mathcal{D}} \left\|l_i^{(k)}\right\|_2^2 = 1 \tag{46}$$

Finally, we define $\tilde{s}_{ij}$, which is an estimator of $s_{ij}$:

$$\hat{s}_{ij} = 1 - \frac{\hat{I}(y_i; \boldsymbol{x}|z_j)}{\hat{I}(y_i; \boldsymbol{x})} \tag{47}$$

$$= 1 - \frac{1}{|D|} \sum_{\left(l_i^{(k)}, z_j^{(k)}\right) \in \mathcal{D}} \left\|l_i^{(k)} - f_{ij}\left(z_j^{(k)}, \phi\right)\right\|_2^2 \tag{48}$$

We note that in Algorithm 2 we use the nomenclature $y_i^{(k)}$ instead of $l_i^{(k)}$ for ease of understanding.

## C  MEASURING FACTORS AND REPRESENTATIONS INVARIANCE AND EXPLICITNESS

In section 4.2 we argue that it is more convenient to measure sufficiency and minimality than the properties of section 3 separately. However, it could be the case that someone wanted to measure them individually. Thus, we propose methods for measuring factors, and representations invariance and explicitness. We also evaluate through these metrics the scenarios presented in section 5.1.

### C.1  CALCULATION OF FACTORS-INVARIANCE

As we have explained, a variable $z_j$ is factors-invariance with respect to $y_i$ when $I(z_j; \boldsymbol{y}|y_i) = I(z_j; \boldsymbol{y}) - I(z_j; y_i) = 0$. We would like our metric for factors-invariance to be in the range $[0, 1]$. Therefore, we define the factors-invariance of $z_j$ with respect to $y_i$:

$$FI_{ij} = 1 - \frac{I(z_j; \boldsymbol{y}|y_i)}{I(z_j; \boldsymbol{y})} = \frac{I(z_j; y_i)}{I(z_j; \boldsymbol{y})} \tag{49}$$

i.e., $FI_{ij}$ collects the proportion of information about $z_j$ present in $y_i$ relative to that present in $\boldsymbol{y}$. Although it is useful to analyze $FI_{ij}$ to obtain information about what is the connection between each factor and variable, it is also interesting to have a single term that determines how factors-invariant a representation is. We define this term below:

$$\bar{FI} = \frac{1}{n_z} \sum_{j=1}^{n_z} \max_i FI_{ij} \tag{50}$$

Unlike minimality and sufficiency, here we do have estimators in the literature for $I(z_j; y)$ and $I(z_j; y_i)$ Kozachenko & Leonenko (1987); Paninski (2003). Therefore, we could either use one of these estimators or obtain an approximation following a process analogous to those described in Appendix B. In Algorithm 3 we give a way to obtain $\bar{FI}$ following the latter approximation to maintain consistency with section 4.3. In this algorithm we construct two regressors or classifiers to predict $z_j$: one from $\boldsymbol{y}$ and one from $y_i$. We know that a representation is factors-invariant if $p(z_j|y_i) = p(z_j|\boldsymbol{y})$. In the following we give an intuitive explanation of the results to be expected. On the one hand, when $y_i$ contains all the information about $z_j$, then the predictions from $y_i$ and $\boldsymbol{y}$ will be equal and $\hat{FI}_{ij} = 1$. On the other hand, if $y_i$ contains little information about $z_j$, but other factors of $\boldsymbol{y}$ do contain information about $z_j$, then $\hat{FI}_{ij} < 1$. In this case, the more information about $z_j$ there is in $\boldsymbol{y}$, the smaller $\hat{FI}_{ij}$ will be. Finally, if $z_j$ contains no information about $bmy$, then $\hat{FI}_{ij} = 1$. That is, we are not measuring nuisances-invariance. This extreme case presents the importance of measuring nuisances-invariance in conjunction with factors-invariance (via minimality), since $\hat{FI}_{ij}$ can be maximal even in the case where $z_j$ contains no information about any factors.

---

**Algorithm 3** Calculation of Factors-Invariance

**Input:** $\left\{ \{y_i^{(k)}\}_{i=1}^n \right\}, \left\{ \{z_j^{(k)}\}_{j=1}^m \right\}$
**Output:** $\bar{FI}$
1: $\{y_i^{(k)}\} \leftarrow \text{StandardScale}(\{y_i^{(k)}\})$, $i = 1$ to $n$
2: $\{z_j^{(k)}\} \leftarrow \text{StandardScale}(\{z_j^{(k)}\})$, $j = 1$ to $m$
3: **for** $j = 1$ to $m$ **do**
4:     $f_j \leftarrow \text{fit}\left( \{\boldsymbol{y}^{(k)}\}, \{z_j^{(k)}\} \right)$
5:     **for** $i = 1$ to $n$ **do**
6:         $f_{ij} \leftarrow \text{fit}\left( \{y_i^{(k)}\}, \{z_j^{(k)}\} \right)$
7:         $\hat{FI}_{ij} \leftarrow 1 - \frac{1}{K} \sum_k \left\| f_{ij}\left( y_i^{(k)} \right) - f_j\left( \boldsymbol{y}^{(k)} \right) \right\|_2^2$
8:     **end for**
9: **end for**
10: $\bar{FI} \leftarrow \frac{1}{m} \sum_{j=1}^n \max_i(\hat{FI}_{ij})$

---

In Figure 7 we show the values of factors-invariance in the first and third experiments of section 5.1, since these are the experiments designed to modify the value of factors-invariance. First, we see that when $\beta$ takes low values (i.e., the representation is nuisances-invariant), the metric is sensitive to the level of factors-invariance (equivalently, it varies according to the value of $\alpha$). However, as the values of $\beta$ go up, then the metric becomes insensitive to the factors-invariance level, as we discussed in the previous paragraph. Therefore, if we were using a metric to measure only the level of factors-invariance, it would not be reliable in the presence of nuisances in the input. It should also be noted that in the figure 7b the value is maximum when $\alpha = 0$ regardless of the dependence value of the factors (i.e., it is independent of $\delta$), which means that it works correctly when the factors are correlated.

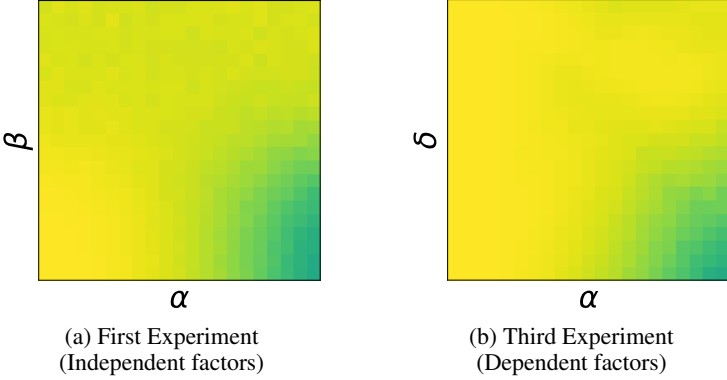

(a) First Experiment
(Independent factors)

(b) Third Experiment
(Dependent factors)

Figure 7: Values of Factors-Invariance in the first and third scenarios of section 5.1. We note that $\alpha \in [0, 0.5]$, $\beta \in [0, 1]$ and $\delta \in [0, 0.5]$ and the values of the metrics go from 0 to 1.

## C.2   CALCULATION OF REPRESENTATIONS-INVARIANCE

As we have explained, a variable $z_j$ is factors-invariance with respect to $y_i$ when $I(y_i; \boldsymbol{z}|z_j) = I(y_i, \boldsymbol{z}) - I(y_i; z_j) = 0$. We would like our metric for representations-invariance to be in the range $[0, 1]$. Therefore, we define the factors-invariance of $z_j$ with respect to $y_i$:

$$RI_{ij} = 1 - \frac{I(y_i; \boldsymbol{z}|z_j)}{I(y_i; \boldsymbol{z})} = \frac{I(y_i; z_j)}{I(y_i; \boldsymbol{z})} \tag{51}$$

i.e., $RI_{ij}$ collects the proportion of information about $y_i$ present in $z_j$ relative to that present in $\boldsymbol{z}$. Although it is useful to analyze $RI_{ij}$ to obtain information about what is the connection between each factor and variable, it is also interesting to have a single term that determines how factors-invariant a representation is. We define this term below:

$$\bar{RI} = \frac{1}{n_y} \sum_{i=1}^{n_y} \max_j RI_{ij} \tag{52}$$

Again we have estimators in the literature for $I(y_i; \boldsymbol{z})$ and $I(y_i; z_j)$, which we could use to estimate this term. In Algorithm 4, we give a method to estimate $\bar{RI}$, in which we have followed a process analogous to the one presented in Appendix B to maintain consistency with section 4.3. In Algorithm 4 we construct two regressors or classifiers to predict $y_i$: one from $\boldsymbol{z}$ and one from $z_j$. We know that a representation is representations-invariant if $p(y_i|z_j) = p(y_i|\boldsymbol{z})$, so intuitively we have the following. When $z_j$ contains all information about $y_i$, then the predictions from $z_j$ and $\boldsymbol{z}$ will be equal and $\hat{RI}_{ij} = 1$. On the other hand, if $z_j$ contains little information about $y_i$, but other variables from $\boldsymbol{z}$ do contain information about $y_i$, then $\hat{RI}_{ij} < 1$. In this case, the more information about $y_i$ in $\boldsymbol{z}$, the smaller $\hat{RI}_{ij}$ will be. Finally if $\boldsymbol{z}$ contains no information about $y_i$, then $\hat{RI}_{ij} = 1$, this metric would take its maximum value even when $\boldsymbol{z}$ contains no information about $y_i$. This extreme case demonstrates the importance of measuring representations-invariance in conjunction with explicitness (via minimality), since $\hat{FI}_{ij} < 1$ can be maximal even in the case where $z_j$ does not contain any information about $y_i$.

---

**Algorithm 4** Calculation of Representations-Invariance

---

**Input:** $\left\{ \{y_i^{(k)}\}_{i=1}^n \right\}, \left\{ \{z_j^{(k)}\}_{j=1}^m \right\}$
**Output:** $\bar{RI}$
1: $\{y_i^{(k)}\} \leftarrow \text{StandardScale}(\{y_i^{(k)}\})$, $i = 1$ to $n$
2: $\{z_j^{(k)}\} \leftarrow \text{StandardScale}(\{z_j^{(k)}\})$, $j = 1$ to $m$
3: **for** $i = 1$ to $n$ **do**
4:     $f_i \leftarrow \text{fit}\left( \{\boldsymbol{z}^{(k)}\}, \{y_i^{(k)}\} \right)$
5:     **for** $j = 1$ to $n$ **do**
6:         $f_{ij} \leftarrow \text{fit}\left( \{z_j^{(k)}\}, \{y_i^{(k)}\} \right)$
7:         $\hat{RI}_{ij} \leftarrow 1 - \frac{1}{K} \sum_k \left\| f_{ij}\left( z_j^{(k)} \right) - f_i\left( \boldsymbol{z}^{(k)} \right) \right\|_2^2$
8:     **end for**
9: **end for**
10: $\bar{RI} \leftarrow \frac{1}{n} \sum_{i=1}^n \max_j(\hat{RI}_{ij})$

---

In Figure 8 we show the values of representations-invariance in the second and fourth experiments of section 5.1, since these are the experiments designed to modify the value of representations-invariance. First, we see that when $\gamma$ takes low values (i.e., the representation is explicit), the metric is sensitive to the level of representations-invariance (equivalently, it varies according to the value of $\alpha$). However, as the values of $\gamma$ go up, then the metric becomes insensitive to the representations-invariance level, as we described in the previous paragraph. Therefore, a metric that measures only the level of representations-invariance independently of the level of explicitness is not reliable, since these two properties are related. It should also be noted that in the figure 8b the value is maximum when $\alpha = 0$ regardless of the dependence value of the factors (i.e., it is independent of $\delta$), so it works correctly when the factors are correlated.

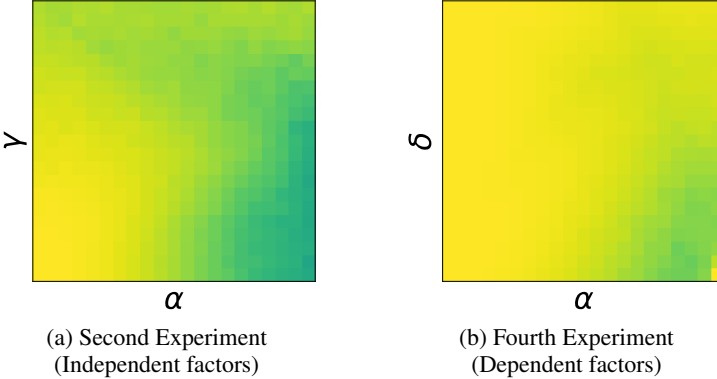

(a) Second Experiment
(Independent factors)

(b) Fourth Experiment
(Dependent factors)

Figure 8: Values of Factors-Invariance in the second and fourth scenarios of section 5.1. We note that $\alpha \in [0, 0.5]$, $\gamma \in [1, 2]$ and $\delta \in [0, 0.5]$ and the values of the metrics go from 0 to 1.

### C.3 CALCULATION OF EXPLICITNESS

Finally, we have defined an explicit representation as that in which $\boldsymbol{z}$ satisfies that $I(y_i; \boldsymbol{x}|\boldsymbol{z}) = I(y_i, \boldsymbol{z}) - I(y_i; \boldsymbol{x}) = 0$. We would like our metric for explicitness to be in the range $[0, 1]$. Therefore, we define the explicitness of $\boldsymbol{z}$ with respect to $y_i$:

$$E_i = 1 - \frac{I(y_i; \boldsymbol{x}|\boldsymbol{z})}{I(y_i; \boldsymbol{x})} = \frac{I(y_i; \boldsymbol{z})}{I(y_i; \boldsymbol{x})} \tag{53}$$

i.e., $E_i$ collects the proportion of information about $y_i$ present in $\boldsymbol{z}$ relative to that present in $\boldsymbol{x}$. To calculate the level of the explicitness of a variable, we just simply calculate the mean as:

$$\bar{E} = \frac{1}{n_y} \sum_{i=1}^{n_y} E_i \tag{54}$$

In this case we do not have good estimators for $I(y_i; \boldsymbol{x})$. However, we can follow a process analogous to that of sufficiency estimation based on Appendix B. In Algorithm 5 we construct a regressor or classifier to predict $y_i$: one from $\boldsymbol{z}$. We know that a representation is explicit if $p(y_i|\boldsymbol{z}) = p(y_i|\boldsymbol{x})$. Intuitively, we can see that the more information $\boldsymbol{z}$ contains about $y_i$, the better the predictions of $f_i$ will be and hence $E_i$ will be higher. However, with this metric we are not evaluating how the different factors separate in the representation.

---

**Algorithm 5** Calculation of Explicitness

**Input:** $\left\{ \{y_i^{(k)}\}_{i=1}^n \right\}, \left\{ \{z_j^{(k)}\}_{j=1}^m \right\}$
**Output:** $\bar{E}$
1: $\{y_i^{(k)}\} \leftarrow$ StandardScale($\{y_i^{(k)}\}$), $i = 1$ to $n$
2: $\{z_j^{(k)}\} \leftarrow$ StandardScale($\{z_j^{(k)}\}$), $j = 1$ to $m$
3: **for** $i = 1$ to $n$ **do**
4: $\quad f_i \leftarrow$ fit $\left( \{\boldsymbol{z}^{(k)}\}, \{y_i^{(k)}\} \right)$
5: $\quad \hat{E}_i \leftarrow 1 - \frac{1}{K} \sum_k \left\| y_i^{(k} - f_i\left(\boldsymbol{z}^{(k)}\right) \right\|_2^2$
6: **end for**
7: $\bar{E} \leftarrow \frac{1}{n} \sum_{i=1}^n (\hat{E}_i)$

---

In Figure 9 we show the explicitness values in the second experiment of section 5.1, which is the case designed to modify the explicitness level. First, we see that when $\alpha$ takes low and medium values (i.e., the representation has a representations-invariant level), the metric is sensitive to the level of explicitness (equivalently, it varies according to the value of $\gamma$). However, for higher values of $\alpha$, then the metric gradually becomes more insensitive to the level of explicitness. This may be because the classifier or regressor $f_i$ does not have sufficient complexity to obtain $y_i$ when $\boldsymbol{z}$ has a high level of representations-invariance.

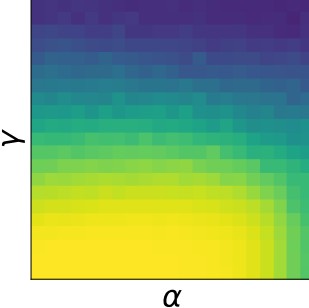

Figure 9: Values of Factors-Invariance in the second scenario of section 5.1. We note that $\alpha \in [0, 0.5]$, $\gamma \in [1, 2]$ and $\delta \in [0, 0.5]$ and the values of the metrics go from 0 to 1.

# D  MORE RESULTS IN LITERATURE DATASETS EXPERIMENT

In this Appendix, we give similar results to those of section 5.2 but for Shapes3D, DSprites and CelebA. The conclusions that can be drawn from these datasets are similar to those for MPI3D. Thus, description of section 5.2 applies also here. The purpose of these figures is to illustrate that our metrics present similar behaviors for different datasets. We note that it is not possible to have three pairs for DSprites since it only has 5 factors of variation.

## D.1  MINIMALITY RESULTS

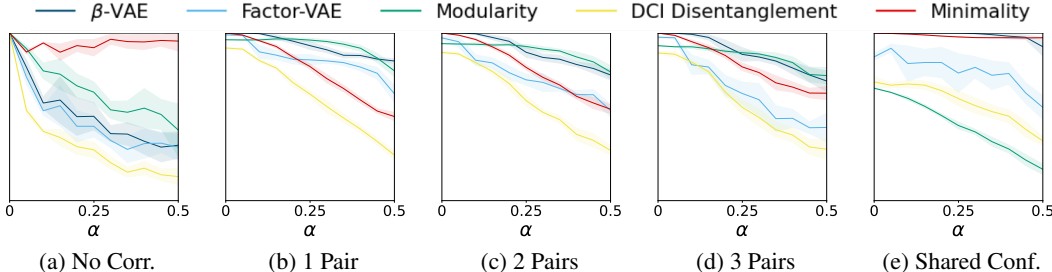

(a) No Corr.   (b) 1 Pair   (c) 2 Pairs   (d) 3 Pairs   (e) Shared Conf.

Figure 10: Metrics measuring minimality for Shapes3D. The values of all the metrics go from 0 to 1.

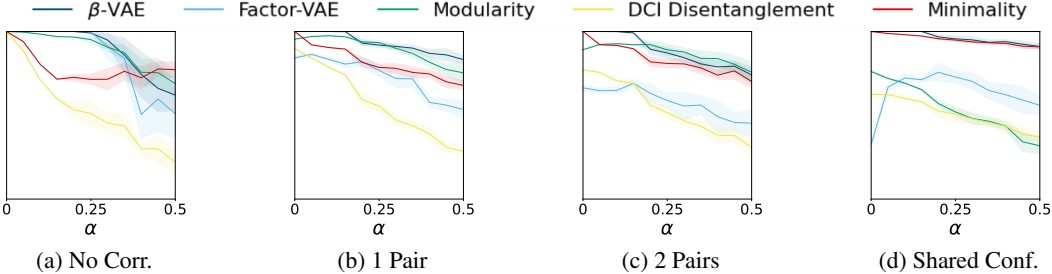

(a) No Corr.   (b) 1 Pair   (c) 2 Pairs   (d) Shared Conf.

Figure 11: Metrics measuring minimality for DSprites. The values of all the metrics go from 0 to 1.

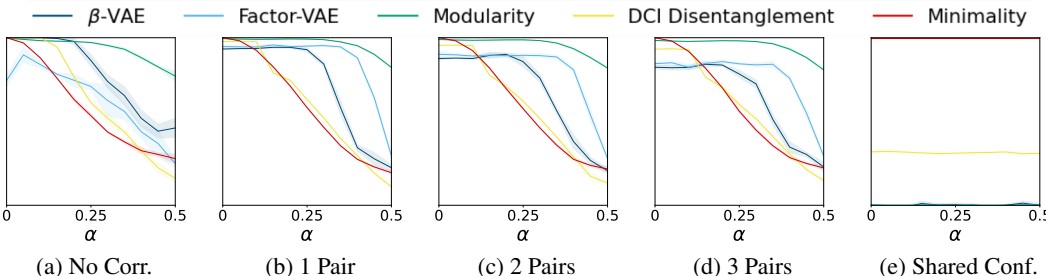

(a) No Corr.   (b) 1 Pair   (c) 2 Pairs   (d) 3 Pairs   (e) Shared Conf.

Figure 12: Metrics measuring minimality for CelebA. The values of all the metrics go from 0 to 1.

## D.2  SUFFICIENCY RESULTS

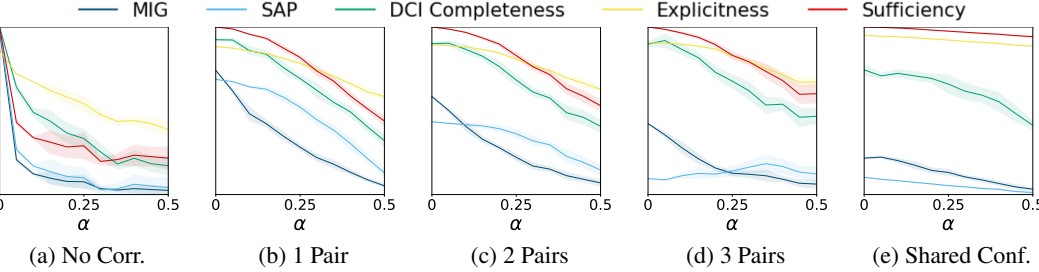

(a) No Corr.   (b) 1 Pair   (c) 2 Pairs   (d) 3 Pairs   (e) Shared Conf.

Figure 13: Metrics measuring sufficiency for Shapes3D. The values of all the metrics go from 0 to 1.

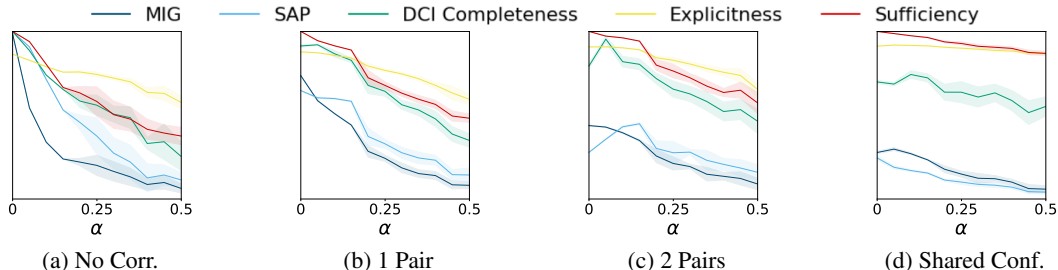

(a) No Corr.  (b) 1 Pair  (c) 2 Pairs  (d) Shared Conf.

Figure 14: Metrics measuring sufficiency for DSprites The values of all the metrics go from 0 to 1.

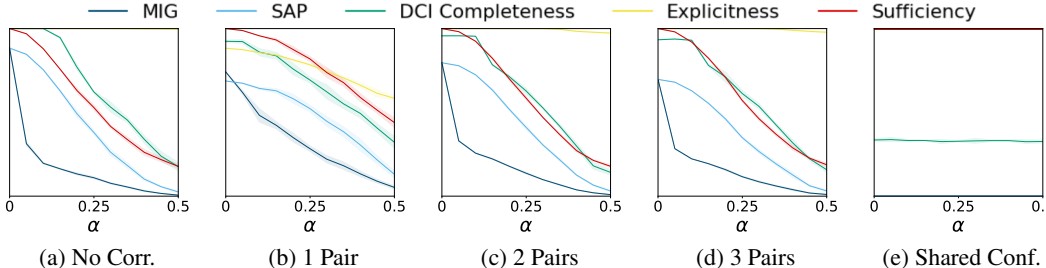

(a) No Corr.  (b) 1 Pair  (c) 2 Pairs  (d) 3 Pairs  (e) Shared Conf.

Figure 15: Metrics measuring sufficiency for CelebA. The values of all the metrics go from 0 to 1.

# E MORE DETAILS ABOUT LITERATURE DATASETS EXPERIMENT

We show next what pairs are correlated in experiment in section 5.2. All the correlations are with $\sigma = 0.1$, i.e., they are strong. The reason for this is that we want to analyze the performance of different metrics under strong correlation scenario. We provide multiple combinations of pairs per scenario (e.g. in MPI3D and 1 Pair, we give three combinations). We calculate the value of the metrics in each combination and obtain the mean of all of them, which are the results provided.

Table 1: Correlation in MPI3D Dataset

| | |
|---|---|
| **1 Pair** | • {cameraHeight, backgroundCol}
• {objCol, objSize}
• {posX, posY} |
| **2 Pairs** | • {objCol, objShape}, {posX, posY}
• {objCol, posX}, {objShape, posY} |
| **3 Pairs** | • {objCol, backgroundCol}, {cameraHeight, posX}, {objShape, posY}
• {objCol, posX}, {objShape, posY}, {backgroundCol, cameraHeight} |
| **Shared Conf.** | • objShape against all other factors.
• posX against all other factors. |

Table 2: Correlation in Shapes3D Dataset

| | |
|---|---|
| **1 Pair** | • {floorCol, wallCol}
• {objType, objSize}
• {objType, wallCol}
• {objType, objCol} |
| **2 Pairs** | • {objSize, floorCol}, {objType, wallCol}
• {objSize, objType}, {floorCol, wallCol}
• {objType, objCol}, {floorCol, wallCol} |
| **3 Pairs** | • {objSize, objAzimuth}, {objType, wallCol}, {floorCol, objCol}
• {objCol, objAzimuth}, {objType, objSize}, {floorCol, wallCol} |
| **Shared Conf.** | • objSize against all other factors.
• wallCol against all other factors. |

Table 3: Correlation in DSprites Dataset

| | |
|---|---|
| **1 Pair** | • {shape, scale}
• {posX, posY}
• {shape, posY} |
| **2 Pairs** | • {shape, scale}, {posX, posY}
• {shape, psoX}, {scale, posY} |
| **Shared Conf.** | • shape against all other factors.
• posX against all other factors. |

Table 4: Correlation in CelebA Dataset

| | |
|---|---|
| **1 Pair** | • {bagsUnderEyes, bald}
• {5OClockShadow, attractive}
• {bangs, bigLips} |
| **2 Pairs** | • {5OClockShadow, archedEyebrows}, {bangs, bigLips}
• {5OClockShadow, bangs}, {archedEyebrows, bigLips} |
| **3 Pairs** | • {5OClockShadow, bald}, {bagsUnderEyes, bangs}, {archedEyebrows, bigLips}
• {5OClockShadow, bangs}, {objType, objSize}, {archedEyebrows, bigLips} |
| **Shared Conf.** | • archedEyebrows against all other factors.
• bangs against all other factors. |

