# OpenReview forum: "Defining and Measuring Disentanglement for non-Independent Factors of Variation"
_ICLR.cc/2025/Conference — Submitted to ICLR 2025_

### Official Review · Reviewer_k1fY · 2024-10-30

**Soundness:** 2
**Presentation:** 3
**Contribution:** 2
**Rating:** 5
**Confidence:** 3

**Summary:**

The authors propose measuring disentanglement through Minimality and Sufficiency.
They propose two metrics based on upper bounds of the before mentioned quantities.
They demonstrate their metrics on synthetic experiments. They claim that their metrics are better suited for correlated datasets.

**Strengths:**

The paper is well written in general. The introduction and related work section position the reader adequately. Using the concept of minimality and sufficiency is thought provoking.

**Weaknesses:**

- If I understand correctly, the authors decided to use an upper bound for their metric, but this design choice lacks motivation and is only noted in the Appendix.

- The experimental section is, unfortunately, not convincing to me. Multiple design choices of the authors are not motivated. The authors did not investigate their metrics's correlation with down-stream task utility.
The authors never looked at actually learned representations, but only used simple linear (and in the first section, trigonometric) mappings. The discussion the authors give is mostly limited to extreme/edge cases of their metric. The metrics overall behaviour is not discussed.

**Questions:**

- Appendix B.1 is titled *DERIVATION OF THE ESTIMATOR FOR MINIMALITY* and B.2 is titled *DERIVATION OF THE ESTIMATOR FOR SUFFICIENCY.* However, the authors seem to derive upper bounds. Did I understand this correctly? If so, how tight is the bound? Can the authors please elaborate why they believe an upper bound is sufficient for a metric?

- In Section 4.3 The authors introduce $\bar{m}$ and $\bar{s}$, writing "it is also interesting to have a single term that determines how minimal a representation z is."
Why did the authors decide to go with the max here? Why not use some other kind of aggregation? Are the  $\bar{m}$, $\bar{s}$ metrics the ones used in the experimental section?

- In the experimental section 5.2, the authors generate a mapping $A \in \mathcal{R}^{n_y \times n_y}$ by fixing the diagonal values to $1- \alpha $ and sample the non-diagonal values from a uniform distribution between 0 and $\alpha$.
Did I understand this correctly? Would then $\bar{m} = \frac{1}{n_y} \sum_{j=1}^{n_y}m_{jj}$ and $\bar{s} = \frac{1}{n_y} \sum_{j=1}^{n_y}s_{jj}$ for $\alpha < 0.5$ ? If so, why do we see a decrease for $0 < 0.5 < \alpha$ in (a), (b) and (c) of Figure 5 but not in (d) (e) and (f)?

- Why is the disentanglement metric DCI-D at 0 for dSprites and $\alpha = 0$?

- In Appendix D, Figure 13, 14 and 15 seem to have the same caption? Is this a mistake? Intuitively I must say that the gradient in Sufficiency looks less pronounced than for other metrics (e.g. Disentanglement). Can the Authors please elaborate on the plots in Appendix D, how do they show the superiority of their metric?

- No standard deviations are provided in Figures 5 and 6 hinting at a single run, which, given the dynamics of the curves, seems insufficient. Either the authors add std's, or they move one of the Figures of Appendix D into the main paper, as it depicts many more runs.

---

> ### Author Response · Authors · 2024-11-23
> **Answer to Reviewer k1fY**
>
> We would like to start by thanking you for your review. Next we try to address each of your questions and concerns. Next bullet points follow the same order as yours in Weaknesses section:
>
> * Actually we do not propose a upper bound but an approximation. During the development of this work, our first goal was to propose a upper bound but we finally found a stronger metric, which was an approximation, that is the current one. We forgot to change some of the sentences in the demonstrations in the Appendices. Please see the new version. Thanks for noticing this point.
> * This second point refers to different concerns related to the experiments section. We agree with the last one but we disagree with the others. We explain why below:
>     1. "Multiple design choices of the authors are not motivated.". We would appreciate it if you could specify better what are the design choices that we do not motivate. The only design choice that we make is that of defining the datasets and we believe that they are sufficiently motivated. We would thank you if you could be more specific with this point.
>     2. "The authors did not investigate their metrics's correlation with down-stream task utility.". Please, find Point 2 in Answer to Reviewer M4R5 II.
>     3. "The authors never looked at actually learned representations, but only used simple linear (and in the first section, trigonometric) mappings.". The reason for this is simply that if we analyze some learned representations, we cannot know if these representations are actually disentangled.
>         In fact, our metric's goal is to analyze if learned representations are disentangled. Thus, we cannot analyze how good the performance of our metric is through learned representations since we would be falling into a _circulus in probando_, which is a logical fallacy. Reviewer M4R5 also pointed out the inconvenience of using learned representations for our proposal.
>     4. "The discussion the authors give is mostly limited to extreme/edge cases of their metric. The metrics overall behaviour is not discussed.". See new Experiment 5.2.
>
> We answer next the Questions following the same order as yours:
> * Please find the answer in the first point of the previous list.
> * For the case of minimality, the reason for choosing the maximum is that we assume that the factor "associated" to a variable $j$ corresponds to the $\arg\max_{i} m_{ij}$. Thus, we want to know how much information of $z_j$ contains its "associated" factor variation $y_i$. Same reasoning can be applied to sufficiency.
> * See "Answer to All Reviewers".
> * See "Answer to All Reviewers".
> * See "Answer to All Reviewers".
> * See "Answer to All Reviewers".
>
> We hope that we have clarified your questions and strengthen our weaknesses. If so and you believe that current version of the work deserves an increase in the score, we would appreciate it if you make that increase effective. Otherwise, we will be willing to continue with the discussion.

---

> > ### Comment · Reviewer_k1fY · 2024-11-25
> > **Thank you for your rebuttal.**
> >
> > Dear Authors,
> > thank you for your rebuttal and for adding standard deviations to your plot.
> >
> > - I do not see how my third question was answered in the "Answer to All Reviewers" section. Can you please elaborate.
> > - I do not see how my fourth question was answered in the "Answer to All Reviewers" section. Can you please elaborate.
> > - I do not see how my fifth question was answered in the "Answer to All Reviewers" section. Can you please elaborate.
> >
> > I still believe, that this paper would benefit from showcasing (not validating) the metric on actual learned representations. Showcasing your metric on actual learned representations of popular disentanglement datasets and analysing the dimensions identified by your metric (e.g. visually) would be convincing and showcase actual utility for the community.
> >
> > I am still sceptic of the authors approach of using the maximum in their metric. Papers such as [1] have showcased, that often multiple dimensions have very similar contributions to one generative factor. Also in the synthetic experiments the authors propose themselves, the contributions of multiple dimensions can be very similar. The "winner takes it all"-approach makes a metric behave more erratic, tiny differences in how much a dimension encodes can then alter the metric dramatically.
> >
> > [1] Eastwood, Cian, and Christopher KI Williams. "A framework for the quantitative evaluation of disentangled representations." 6th International Conference on Learning Representations. 2018.

---

> > > ### Author Response · Authors · 2024-11-30
> > > **Inclusion of actual neural representations and downstream tasks**
> > >
> > > Thank you again for your review. Please find in the general comment the modifications done in experiment 5.2. Now (i) we provide actual neural representations; (ii) we compare them to performance in downstream tasks; and, (iii) if the paper is accepted, we will include an appendix with some visualizations to motivate the contribution and understanding of our work.
> > >
> > > If these last changes served to improve your opinion on the paper and you believe that it deserves a score raise, we would appreciate if you made that raise effective.

---

> ### Author Response · Authors · 2024-11-25
> **Thank you for your quick response.**
>
> Thank you for your quick response. We answer below your comments.
>
> * In the new experiment, we do not contemplate the case in which $0.5 < \alpha$ because it can become hard to interpret. If $0.5 < \alpha$, then there is not a factor of variation that is always more important than others, conclusions are harder to extract.
> * We have slightly modified the experiment 5.2 and we do not find this phenomenon now. We have not found any reason why this was happening in the experiment in the previous version of our manuscript.
> * Figures 13, 14 and 15 are now modified (in fact, paper no longer has figures 14 and 15). In explanation of new section 5.2 we give the reason why _Sufficiency_ takes the values that it takes and the reason why this is a desirable behaviour.
>
> > I still believe, that this paper would benefit from showcasing (not validating) the metric on actual learned representations. Showcasing your metric on actual learned representations of popular disentanglement datasets and analysing the dimensions identified by your metric (e.g. visually) would be convincing and showcase actual utility for the community.
>
> We do not think that obtaining representations from data and just showing the values of our metrics for these representations helps to make the paper more convincing.
> However, maybe the next experiment can help to understand the necessity of our proposal: Train a variation of a VAE that tries to learn disentangled representations for a dataset with correlated factors of variation (e.g. one of those used in Experiment 5.2). Then, show that we can generate data in a controlled way (and thus, the representations are presumably disentangled) and that, despite of this, other metrics in the literature indicate that the representations are not disentangled.
> If you believe that this experiment can help to showcasing our metrics, we can include it in an appendix in the final version of the paper. However we do not believe that this must be in the main text since, as explained in Point 2 in Answer to Reviewer M4R5 II, we think that measuring disentanglement performance through the performance in a downstream task is not methodologically correct.
>
> > I am still sceptic of the authors approach of using the maximum in their metric. Papers such as [1] have showcased, that often multiple dimensions have very similar contributions to one generative factor. Also in the synthetic experiments the authors propose themselves, the contributions of multiple dimensions can be very similar. The "winner takes it all"-approach makes a metric behave more erratic, tiny differences in how much a dimension encodes can then alter the metric dramatically.
>
> This is exactly the goal of our paper. We want to propose a method that allows to different dimensions of the representation to have some information about a factor of variation $y_i$. The point is that in some of the dimensions this information must come through another factor of variation different than $y_i$ (but that is correlated with $y_i$). This is the concept of conditional independence, and this is why we propose using conditional mutual information rather than mutual information to measure disentanglement. In section 3 we argue why this is desirable and in section 4 we demonstrate that our metrics are actually (almost) measuring this.

---

### Official Review · Reviewer_M4R5 · 2024-11-02

**Soundness:** 2
**Presentation:** 3
**Contribution:** 3
**Rating:** 5
**Confidence:** 4

**Summary:**

This paper proposes to define and measure disentanglement in a way that is more relevant for real-world scenarios, where (1) the true generative factors of variation are not necessarily independent, and (2) there are nuisance factors which are not relevant for a given task but may act as confounders. The metrics proposed in this paper are somewhat similar to those proposed by Ridgeway & Mozer (2018) and Eastwood & Williams (2018), but generalized/extended in the 2 ways mentioned above. Factors are still assumed to be independendent here, but only given the observed raw data, while in previous work they were unconditionally independent.

The authors consider 4 properties:
1. Factors-invariance: $z_j$ is factors-invariant for the factor $y_i$ if, given $y_i$, there is no mutual information between $z_j$ and all the other factors in $\mathbf{y}$. This is a direct extension of modularity/disentanglement that accounts for correlations.
2. Nuisances-invariance: this is the same as factors-invariances, except we replace "all the other factors" with "all nuisance variables".
3. Representations-invariance: $z_j$ is representations-invariant for the factor $y_i$ if, given $z_j$, there is no mutual information between $y_i$ and all the other representations in $\mathbf{z}$. This is a direct extension of compactness/completeness that accounts for correlations.
4. Explicitness: $\mathbf{z}$ is explicit for the factor $y_i$ if, given the full representation $\mathbf{z}$, the data $\mathbf{x}$ provides no additional information about $y_i$, i.e., $y_i$ is fully described by $\mathbf{z}$.

The authors then show that:
- Minimality is equivalent to (1) and (2) jointly.
- Sufficiency is equivalent to (3) and (4) jointly.

They also argue that it is reasonable to focus on minimality and sufficiency, and propose methods to estimate these quantities in practice, and showcase their metrics alongside classic disentanglement metrics in a few toy experiments.

**Strengths:**

- Very well-written introduction, with a good explanation of the motivation. In general the background and setup are nicely laid out, which is a service to the community.
- Overall pretty easy to follow, with clear and convincing arguments in the background/theory section.
- Interesting ideas that are relevant for the community.

**Weaknesses:**

**Clarification on minimality and sufficicency.** As far as I understand, minimality and sufficiency are always defined w.r.t. a task, and considering the entire representation. Here, the authors talk about "task" to distinguish between relevant factors $\mathbf{y}$ and nuisances $\mathbf{n}$. This is consistent with previous work on representation learning that concerns minimality and sufficiency. However, in this paper, the 4 properties (factors-invariance etc.) are linked to minimality and sufficiency in a different context, where the representations $z_j$ are considered separately, and the "tasks" are the single factors $y_i$. Is my interpretation correct? Regardless, I think this point should be clarified and made more explicit, to avoid potential misunderstandings. (This could also help give a very precise link between the 4 properties here and the classic disentanglement metrics by Ridgeway & Mozer (2018) and Eastwood & Williams (2018))

**Beginning of Sec. 5:**
> In this section we use minimality and sufficiency (lowercase) to refer to the properties of section 3 and Minimality and Sufficiency (uppercase) to refer to the metrics of section 4.3.

Two issues here:
1. Minimality and sufficiency are not mentioned in Section 3. If the authors mean to distinguish between properties and metrics, then I guess the reference should be to Section 4.1?
2. Distinguishing only with capitalization sounds like a recipe for misunderstandings. I would suggest using letters/symbols or acronyms, for example.

**Experiments.**

The theoretical arguments for why these metrics make more sense than existing ones are very clear. The empirical arguments, however, less so.

1. On the comparison between e.g. minimality and DCI disentanglement in Sec. 5.1: at least in the uncorrelated case, DCI should be good (especially with low nuisance strength). And in fact, I believe it is. For example, I disagree with "only takes high values when factors-invariance and nuisances-invariance levels are very low" being a problem. Absolute values of metrics are not necessarily as interesting as how they rank different representations. I don't think there's anything inherently more "accurate" or "correct" about the Minimality metric than e.g. Minimality$^2$. What about showing scatter plots and/or computing (rank) correlations between the metrics? Would it give a more complete picture perhaps?

2. My major concern is that, as I wrote above, I think the most interesting aspect is actually how different metrics rank different representations. And the ground truth to make comparisons with, should not be an abstract metric, but rather a concrete evaluation whose relevance/usefulness the community can agree on. So I think it would be important to investigate if these metrics can be useful in practice e.g. for model selection for downstream tasks (including e.g. generalization, fairness, sample efficiency etc.), as done by quite a few disentanglement papers especially around 2019--2021. So a research question could be: when using disentanglement metrics to cheaply select representations for downstream tasks, does our metric yield better results than others according to some evaluation metric?

3. Another concern I have is that sometimes these proposed metric don't seem to do a very good job, especially in the toy image datasets with correlations between factors. E.g. in Fig. 5 DCI disentanglement seems quite bad, but it's arguably better than Minimality on MPI3D (and it's not the only one). The situation is even worse for the sufficiency-like metrics in Fig. 6, where there's no particularly obvious advantage of using Sufficiency (most curves are relatively close to each other). I suspect the issue might be resolved if the authors clarify their interpretation of Figs. 5 and 6. Otherwise, this is clearly a limitation, since it's even in the correlated case, where these methods should shine. So it should be addressed upfront, and ideally there should be a bit more investigation into why this happens, what impact it might have (but see my paragraph above, i.e. to measure actual impact there's more work to be done), whether something can be done to mitigate it.

4. In addition, I think it would be interesting and useful to consider different ways/degrees of mixing as e.g. in [Eastwood et al. (2023)](https://arxiv.org/abs/2210.00364). But that's perhaps more of a side quest and it's more related to explicitness as defined in that paper (see comment on this below under "minor"). I wouldn't prioritize this, although maybe a short discussion in passing could be beneficial.

5. Regarding datasets, why use image datasets (Sec. 5.2) if the images as far as I can see are never used? I agree it doesn't make sense to use images, but then why not generate random low-dimensional data (just the factors) since they just need to be mixed artificially? Again, these datasets would be useful when doing model selection in a practical (toy) setting as in classic disentanglement papers, to see if these metrics can indeed be more useful than previous ones.

**Minor**:
- end of page 3: "connected to those described in 3" is a reference to the current section (also, it should be "Section 3" anyway)
- line 286: "since this gap can be low for correlated factors even when zj is minimal." I would maybe expand a bit on this to clarify
- There might be some issues/inconsistencies with notation. As far as I understand, $y_j$ is a factor, there are $n$ factors, and $\mathbf{y} = \{y_i\}_{i=1}^n$ is the set of factors -- all this for a single data point, because e.g. $y_i$ is not bold. But then when "datasets" appear, it seems that actually $y_i$ was the vector of $i$-th factors from the entire dataset, and when considering a single data point we have the notation $\mathbf{y}^{(k)}$, where $k=1,...,K$ and $K$ is the dataset size. I think this notation should be clarified earlier in the manuscript.
- Explicitness as in Ridgeway & Mozer (2018) is a bit misleading, and I think informativeness is much better (on the other hand, note that compactness in my opinion is more descriptive than completeness). Explicitness is also defined as additional metric by [Eastwood et al. (2023)](https://arxiv.org/abs/2210.00364) where perhaps the term explicitness makes more sense.

**In conclusion**, I think these are very interesting ideas and they are well explained. The writing is also overall good. I think the experimental validation is however rather lacking. Note that I'm not asking for any real-world or larger-scale experiments -- I just think to prove the points the authors want to prove, a wider and better-designed experimental study is necessary.

**Questions:**

See weaknesses.

---

> ### Author Response · Authors · 2024-11-23
> **Answer to Reviewer M4R5 I**
>
> We would like to start by thanking you for your review. We answer each of your concerns and questions in the Weaknesses section following the same structure that you propose:
>
> **Clarification on minimality and sufficiency**
>
> Your interpretation is correct. Although the concepts of sufficient and minimal representation have been used in multiple works to refer to one representation and one task, nothing prevents one from doing it for more than one. Actually, as explained in the second paragraph os section 3, the $z_j \in \\pmb{z}$ can be seen as representations. Equivalently, one can see each factor of variation as tasks (actually the word task is just a generalization for something that you want to learn and potentially solve). Thus, we could say that we want to perfectly "solve" only one factor $y_i$ of variation with only one representation $z_j$. In section 3 we connect [1,2] with the four properties and in section 4.1 we connect the four properties with minimal and sufficient representations. We don't know if you consider this a sufficient connection, but for us it is the most clear way of connecting our paper to the two previous.
>
> **Beginning of Sec. 5**
>
> You are right. We explain our solution for both issues. Please let us know if you find them proper.
> 1. Since the metrics are introduced in section 4.3 (the title of the section starts with the word Metrics) we make this clarification in this section. Concretely, after defining $\bar{m}$ and $\bar{s}$, which are the metrics that we propose.
> 2. Now we use italic for metrics and roman for properties.

---

> ### Author Response · Authors · 2024-11-23
> **Answer to Reviewer M4R5 II**
>
> **Experiments**
>
> We try to address each of your specific concerns below:
> 1. This is an interesting question: "Should a metric have an intrinsic meaning or should it be useful only to compare methods?". Although, we do not have clear answer for this, we believe that the first option is never worse than the second, since the first option implies the second one, while the converse is not necessarily true. In fact, many of the most widely used metrics in machine learning are designed to be interpretable (e.g. accuracy, f1-score, BLEU, MAP).
>     Our metrics have a clear interpretation by definition (and, as a consequence, they also allow to compare representations). For example, minimality captures the proportion of information that a variable $z_j$ contains about a factor $y_i$  with respect to the the input $x$. However, other metrics in the literature (e.g. DCI-D) allow to compare representations but they do lack of an easy interpretation.
>     We could add a rank correlation matrix between different metrics in some of the datasets but we do not believe that this would help to draw any specific conclusion on our metrics. **However, we are open to discuss this point further, since we find it interesting and we do not have a clear answer to it.**
>
> 2. This is also an interesting question. However, we believe that comparing the value of a disentanglement metric with the performance in a subset of downstream tasks is not convenient in a paper proposing a definition metric for disentanglement for several reasons, which we list below:
>     * We believe that defining a way of measuring disentanglement has an intrinsic value. Giving the correlation of disentanglement metric and downstream performance could mask the actual intention of the paper.
>     * It is well known that disentangled representations translate into a better performance in downstream tasks. Thus, if out metric correctly measure disentanglement, this should translate into a good correlation between our metric and downstream tasks.
>     * Most importantly and independently of the aforementioned, we believe that measuring the performance in downstream tasks to asses how good a disentanglement metric is is methodologically incorrect. Imagine that we carry out these experiments and the results are successful (in the sense that a disentanglement metric perfectly correlates with downstream performance). Then, saying that this metric perfectly correlates with disentanglement is a logical fallacy, as we can see next:
>           Let $p\equiv \textit{high disentanglement}$, $q\equiv \textit{high value of the disentanglement metric}$ and $r\equiv \textit{good performance in downstream tasks}$. Thus, we have $p \Rightarrow r$ (since we assume that having highly disentangled representations implies a good performance in downstream tasks) and $q \Leftrightarrow r$ (since we assume that the experiments in downstream tasks are successful). Under this scenario we can infer $p \Rightarrow q$ but not $q \Rightarrow p$, i.e., we know that our metric will be high when the disentanglement is high, but our metric can be high even though the disentanglement is low. This derives from the fact that, to the best of our knowledge, it has not be proven that disentanglement is a necessary condition to have good performance in any set of downstream tasks.
>           Thus, measuring performance in downstream tasks does not allow to obtain a bidirectional conclusion. In fact, to the best of our knowledge, it is hard to find a paper proposing a definition and metric for disentanglement apart from that of DCI-ES that perform some experiments in downstream tasks.
>     **We are also open to discuss further this point since it is also of interest of us.**
>
> 3. See "Answer to All Reviewers".
> 4. We use a wide range of mixing degrees (which varies with $\alpha$, $\beta$, $\gamma$, $\delta$ and $\sigma$) along the different experiments. With respect to the ways of mixing, especially if compare the experiments we do with those of DCI-ES paper, we have the next:
>     * We use their _Noisy Labels_ with a wider range of noise values, and their _Linearly-mixed labels_ with a wide variety of $W$ matrices.
>     * With respect to their _Raw data (pixels)_ and _Others_, we believe that it does not make sense to analyze this case in our paper, since we do not know the "ground-truth" level of disentanglement in these cases. Thus, we cannot know if the results that we would obtain make sense or not.
>
> 5. You are right: actual images are never used, but only the factors of variation. Generating randomly the factors is something we do in section 5.1. We found it interesting to use some correlated factors of variation from existing papers, which is what we do in section 5.2.

---

> ### Author Response · Authors · 2024-11-23
> **Answer to Reviewer M4R5 III**
>
> **Minors**
>
> * It should be "those described in section 2". Thanks for noticing.
> * We extended the sentence.
> * We made some modifications to fix this issue. Please see last paragraph of section 4.3 and Algorithms 1 and 2.
> * This is an interesting point. Although in our paper it is a bit oversimplified to avoid misunderstandings, our opinion comparing the metrics that you mention are the next:
>     * _Explicitness_ ($E$) refers to the existence of a simple (e.g., linear) mapping from the representation to the value of a factor while _Informativeness_ ($I$) refers to the amount of information that a representation captures about the underlying. That is, $E \Rightarrow I$, but $I \centernot\Rightarrow E$. Thus, theoretically, they are different. However in practice, $I$ is calculated also through the use of a mapping from the representation to a factor (due to the intractability of the integrals that calculating mutual information involves). Thus, the only practical difference is the type of mapping used to go from the representation to the factor. In fact, both can be seen as ways of _Predictive_ $\mathcal{V}$_-information_ [3] with different _Predictive Family_ $\mathcal{V}$. Concretely, given the _Predictive Family_ used to calculate the Explicitness $\mathcal{V}_E$ and the _Predictive Family_ used to calculate the Informativeness $\mathcal{V}_I$, we have that $\mathcal{V}_E \subset \mathcal{V}_I$, and this is why $E \Rightarrow I$, but $I \centernot\Rightarrow E$.
>     * As far as we understand them, _Compactness_ and _Completeness_ refer to broadly the same idea: The degree to which each underlying factor is captured by one (or a few) representation variable and there is a metric that has been specifically designed to measure _Completeness_ but not to measure _Compactness_.
>
> Sorry for the long answer, but some of your questions refer to ideas that we find interesting to discuss about.
> We hope that we have clarified your questions and strengthen our weaknesses. If so and you believe that current version of the work deserves an increase in the score, we would appreciate it if you make that increase effective. Otherwise, we will be willing to continue with the discussion.
>
> [1] Eastwood, C., \& Williams, C. K. (2018, May). A framework for the quantitative evaluation of disentangled representations. In 6th International Conference on Learning Representations.
>
> [2] Ridgeway, K., \& Mozer, M. C. (2018). Learning deep disentangled embeddings with the f-statistic loss. Advances in neural information processing systems, 31.
>
> [3] Xu, Y., Zhao, S., Song, J., Stewart, R., \& Ermon, S. (2020). A theory of usable information under computational constraints. arXiv preprint arXiv:2002.10689.

---

> ### Comment · Reviewer_M4R5 · 2024-11-26
>
> Thanks for the extensive and interesting rebuttal and for the clarifications. I'm also glad to hear the authors found some questions interesting to discuss! I still have a few more comments on some points.
>
> **Answers about experiments**
>
> 1. That's a good point and I agree that your metric measures something sensible and well-defined. However, I disagree on your point about "clear interpretation". You write "minimality captures the proportion of information that a variable $z_j$ contains about a factor $y_i$ with respect to the the input $x$", and this is imho not intuitively interpretable. E.g. if the metric is equal to 0.76 I'm not sure what to make of it; I know that if another one is 0.84, the latter is supposedly better than the former, but the absolute values are not interpretable. I believe that, in principle, even if they are not interpretable, they are meaningful since they are theoretically justified. They are an estimate of a mathematical quantity that is sensible to want to quantify. But I would argue against saying that e.g. DCI-D is not meaningful. In the uncorrelated case, it is not surprising that some standard metrics may still work well -- that does not undermine this work at all, and overclaiming does it a disservice. If e.g. DCI-D is (almost) a monotonic function of your metric, then your metric in practice does not buy you anything. You can argue that it measures something more sensible, but that's about it. On the other hand, the story is completely different on the dependent case or with nuisance factors, which are the 2 main motivations of this work. I would focus the claim on these cases.
>
> 2.
> 	- "It is well known that disentangled representations translate into a better performance in downstream tasks": I disagree. Disentangled representations are often presented as something desirable, but it's clear by now that they are not always helpful (except for specific cases like interpretability or controllable generation) -- e.g., it depends on the downstream task, downstream model, whether there's a distribution shift at test-time, the nature of such shift, etc. See for example: the 2 papers by Montero et al. that you even already cite; "Challenging common assumptions" which you also already cite, where there's limited evidence of usefulness; [1] where on a harder downstream task there is better sample efficiency but otherwise there's no clear advantage; [2] where it is highlighted how strongly things depend on the downstream model, and for a simple MLP (as opposed to boosted trees used in previous papers) there is a mild advantage only in terms of OOD generalization; [3] where on a RL task disentanglement doesn't seem to help even OOD.
> 	- In addition, you write "Thus, if our metric correctly measure disentanglement, this should translate into a good correlation between our metric and downstream tasks." Previous works have investigated the empirical usefulness of disentanglement through specific metrics. If you expect this correlation to hold, your statement implies that you are measuring disentanglement in a similar way to previous works, therefore making your work less useful in practice (e.g. another metric would rank representations in a similar order to the one yours would). Again, you have the correlated and nuisance cases working for you, so I would not try to claim too much about the standard uncorrelated case. Some previous metrics already work pretty well there, and that is completely fine.
> 	- I agree. My point about usefulness of the metric was in fact orthogonal: if a metric does not necessarily measure disentanglement perfectly (or rather, we cannot know that for certain) but perfectly correlates with some downstream metric we care about (e.g. OOD accuracy on a few different downstream tasks), then it is in any case useful in practice, and the community should know. But again, I agree that it does not necessarily tell us about disentanglement. (In fact, I would even say that it's not necessarily true that $p$ implies $r$, but that is a side note.)
>
> 3. Thanks, the updated results make sense.
>
> 4. You are right, of course. What I meant to write was in fact to have a wide range of nonlinearly mixed representations e.g. by randomly-initialized invertible non-linear transformations (e.g. a composition of a few normalizing flows). I would find it interesting because it would get a bit closer to realistic scenarios while still in a highly controlled setting. However, I still consider it a relatively minor point and I'm not giving it much weight.
>
> 5. Ok, but it is still quite confusing that the title of Sec. 5.2 is "literature datasets" and the datasets in question are image dataset, but at the same time the images are never used. As far as I can tell, the same exact experiments could be run without even mentioning these datasets. Maybe I'm missing something, but that is at the very least unclear.

---

> ### Comment · Reviewer_M4R5 · 2024-11-26
>
> **Additional comments**
>
> I agree that, in order to thoroughly check that these metrics do what expected in highly controlled settings, it would not be particularly helpful to run experiments on learned representations. However, similarly to k1fY's point, I think using learned representations, even on toy image data, is right now a major missing piece in this work -- such experiments would be useful for showcasing the behaviour of these metrics in scenarios that are slightly closer to real applications (which in a way has always been one of the core motivations for disentangled representations, see e.g. Bengio et al. (2013)).
>
> There's also a related point: Träuble et al. and Roth et al., both cited in the submission, use in fact DCI-C as their primary metric without apparent practical issues. E.g. in Träuble et al. the weakly-supervised approach by Locatello et al. [4] allows to learn disentangled representations even on correlated data, and the DCI-D metric is quite high in those cases, and actually from the violin plots you can see it's often very close to 1. Does this somehow depend on the fact that the representations were learned with neural networks, or on the weak supervision in Träuble et al.? Or maybe on the way and/or degree that the factors were entangled in the representations, or on how precisely they were correlated in the data, or on an interplay of the two?
>
> I guess my point is the following. I fully agree with the authors' theoretical arguments, which I find compelling and significant. However, while I am not doubting their empirical results, I think they address only a very narrow setting, which limits the depth of understanding of the proposed metric.
>
> Finally just another note -- I see the point that investigating the correlation with downstream performance is orthogonal, but this would be highly relevant for the community and the authors are missing an opportunity to have an additional, more pragmatic selling point.
>
> **References**
>
> [1] Van Steenkiste, et al. "Are disentangled representations helpful for abstract visual reasoning?", NeurIPS 2019
>
> [2] Dittadi, et al. "On the Transfer of Disentangled Representations in Realistic Settings", ICLR 2021
>
> [3] Träuble, et al. "The Role of Pretrained Representations for the OOD Generalization of Reinforcement Learning Agents", ICLR 2022
>
> [4] Locatello et al. "Weakly-supervised disentanglement without compromises", ICML 2020

---

> > ### Author Response · Authors · 2024-11-30
> > **Actual representations and Downstream tasks performance**
> >
> > Thank you again for your review. Please find in the general comment the modifications done in experiment 5.2. Now (i) we provide actual neural representations; (ii) we compare them to performance in downstream tasks; and, (iii) if the paper is accepted, we will include an appendix with some visualizations to motivate the contribution and understanding of our work.
> >
> > If these last changes served to improve your opinion on the paper and you believe that it deserves a score raise, we would appreciate if you made that raise effective.

---

> > > ### Comment · Reviewer_M4R5 · 2024-12-03
> > >
> > > Thank you for adding these experiments in such a short time. That is for sure helpful, although it would be useful to have a bit more experimental details and metrics as a sanity check (e.g. what are the distributions of the values of these metrics for different VAE methods?). By experimental details I mean, for example:
> > > - The full grid of experiments including potentially random seeds
> > > - How do you exactly compute correlations between pairs? Regarding this, it might be informative to also include scatter plots -- see seaborn pairplot for example.
> > > - How do you set up the downstream tasks? In some papers I mentioned in a previous comment, it starts to be clear that things potentially change a lot when using small MLPs instead of trees/forests.
> > >
> > > To answer your new general comment:
> > > - What does "rest of the elements of the dataset" mean exactly? The correlations should be measured between the metrics across different models, right? I'm not sure I get what different data points have to do with this. Or by elements do you mean factors of variation?
> > > - "those elements that have high minimality in an uncorrelated case should have a high minimality in a correlated case": But correlations affect the degree of disentanglement or minimality of the learned representations, right? In fact, even random seeds may have a huge impact. So I'm not sure I follow.
> > > - "As explained in different works [2,3,4], a disentangled representation should translate into a more sample efficient downstream predictor": I don't think that's what they say in [3] though. So how does that match the results from your experiments?
> > >
> > > Minor note: If accepted, I would also recommend plotting (e.g. heatmaps or something like that) the importance matrices for the different metrics, as done in a few of the disentanglement papers you already cite. I would just do this for a few models selected at random to dig a bit deeper into what is happening. In addition, regarding sample efficiency, I suggest considering other values between 100 and 10k and e.g. plot accuracy curves.
> > >
> > > Overall, I sincerely appreciate the updates and acknowledge the positive direction taken in the rebuttal. However, substantial work remains to be done, particularly in terms of experiments, visualizations, and providing clear explanations for reproducibility. While the promises made are encouraging, it's unclear if they will be fulfilled if the paper is accepted, and most importantly, it will be impossible for them to actually be peer-reviewed. Nonetheless, I am very open to discussing acceptance with the other reviewers and the AC. Either way, all the work gone into the rebuttal is going to be very useful. Thanks for the interesting discussion!

---

### Official Review · Reviewer_9Exd · 2024-11-03

**Soundness:** 3
**Presentation:** 2
**Contribution:** 2
**Rating:** 5
**Confidence:** 4

**Summary:**

The paper addresses the topic of measuring the degree of disentanglement in learned representations. Existing metrics often rely on the assumption that underlying factors are independent, limiting their effectiveness in real-world applications. To address this, the authors propose two complementing metrics: Minimality and Sufficiency. These metrics enable disentanglement assessment without requiring factor independence. Experimental results indicate that Minimality and Sufficiency can identify the presence or absence of disentanglement in scenarios where previous metrics are ineffective.

**Strengths:**

The authors propose two predictor-based metrics, Minimality and Sufficiency, for assessing disentanglement without assuming factor independence. Existing approaches often rely on this independence assumption, which rarely holds in real-world data, limiting the applicability of such metrics to narrow and often unrealistic cases. Minimality and Sufficiency are introduced as complementary, yet opposing, principles that balance trade-offs in capturing disentanglement. The authors provide formal definitions and proofs grounded in Information Theory. Through experiments, they demonstrate that existing metrics struggle to measure disentanglement effectively when factors are not independent, whereas the proposed metrics perform better in these more realistic settings.

**Weaknesses:**

While the authors argue that Minimality and Sufficiency metrics are more applicable to real-world scenarios due to their independence from factor assumptions, this claim would benefit from validation on complex, real-world datasets. For example, using datasets like CelebA, which include nuanced features and dependencies, could illustrate the metrics' practical advantages.

Minimality and Sufficiency are properties associate with information bottleneck techniques, yet this connection is not addressed. Expanding the related work section to discuss information bottleneck approaches would strengthen the paper’s theoretical foundation by situating the metrics within established work. Including studies that use information bottlenecks for disentanglement would clarify how Minimality and Sufficiency build upon or differ from these techniques.

**Questions:**

1. The experiments involve a relatively small number of factors and simpler representations. Could the authors elaborate on how Minimality and Sufficiency perform in more complex scenarios with larger factor sets and higher-dimensional representations? Are there any known limitations in applying these metrics?

2. While Minimality and Sufficiency appear to offer distinct advantages, are there any scenarios where they might fail to capture disentanglement effectively, or where they might be less reliable than existing metrics?

3. In Section 4.3, the authors mention that Minimality and Sufficiency are better suited for cases where factors are not independent, as they focus on the most influential factors and not comparisons across factors. Could existing methods like DCI be modified to relax the independence assumption? If so, would Minimality and Sufficiency still provide a more effective disentanglement assessment, reinforcing the novelty of proposed metrics?

---

> ### Author Response · Authors · 2024-11-23
> **Answer to Reviewer 9Exd**
>
> We would like to start by thanking you for your review. Next we try propose solutions to strengthen the weaknesses that you pointed out. Each bullet point corresponds to each of your two paragraphs.
>
> * CelebA presents a very low level of correlation between the factors of variation. Thus, other metrics perform reasonably well with this. We have extended Experiment 5.2 to make it more "realistic" and to include different levels of disentanglement. Please see "Answer to All Reviewers".
> * The authors are conscious of the connection between minimal-sufficient representations and the Information Bottleneck (IB) and we appreciate the importance of this concept in representation learning. Thus, we have added a paragraph about the IB in section 2, since we agree that this will help to give a better understanding of our approach and to better connect it to other works. However, as far as we know, ours is the first approach connecting the IB with disentangled representation learning. We would thank you if you could provide us some other works doing so in case you are aware of some.
> \end{itemize}
>
> Next, we try to answer your concrete questions:
> 1. We believe that the only limitation is how $f_{ij}$ are chosen. If these regressors are too simple, maybe it could be hard for them to correctly discover the connections between the factors of variation $y_i$ and the representations $z_j$ even if there is a high mutual information between them.
>     This point is briefly commented now in the last sentence of section 4.3. Apart from this, we believe that, assuming that definitions of section 3 are considered reasonable, our metrics should lack of other limitations than the one corresponding to the capacity of the regressors. Concretely, if we assume that definition Section 3 is correct, we have the next:
> (i) in section 4.1 we demonstrate that this definition is equivalent to having minimal-sufficient representations;
> (ii) and in section 4.3 we simply refer to the definition to minimal-sufficient representations to formulate our metrics.
> 2. As mentioned in the previous answer, we believe that the only limitation of our metrics is the way the regressors are chosen. However, as explained in section 2, the most accepted metrics in the literature [1,2] also make use of the regressors. Thus, they present the same limitation as ours.
> 3. First, if we study some of the most used metrics in the literature (DCI or modularity, compactness and explicitness (MCE)) we find it hard to modify them to relax the independence assumption. We believe that they need of too many changes to be considered a modification of the metrics and that they should be considered, if anything, a new metric. Even if our previous affirmation is wrong and we could straightforwardly modify DCI or MCE to make them robust to non-independent factors of variation, the previous metrics would not measure nuisances-invariance, which is a pivotal property that is overlooked in all the methods reviewed in section 2 and that our metric captures.
>
> We hope that we have clarified your questions and strengthen our weaknesses. If so and you believe that current version of the work deserves an increase in the score, we would appreciate it if you make that increase effective. Otherwise, we will be willing to continue with the discussion.
>
> [1] Eastwood, C., \& Williams, C. K. (2018, May). A framework for the quantitative evaluation of disentangled representations. In 6th International Conference on Learning Representations.
>
> [2] Ridgeway, K., \& Mozer, M. C. (2018). Learning deep disentangled embeddings with the f-statistic loss. Advances in neural information processing systems, 31.

---

> > ### Comment · Reviewer_9Exd · 2024-11-25
> >
> > Thank you for extending the experimental results.
> >
> > > CelebA presents a very low level of correlation between the factors of variation.
> >
> > I'm a bit surprised by this; generally, for the less 'extensive' datasets to explore disentanglement in, CelebA is known to be the case where one _cannot_ assume independence between factors (e.g. beard vs gender, etc). It would be useful to get a handle on how much correlation would be required for this method to be useful then?
> >
> > > [...] as far as we know, ours is the first approach connecting the IB with disentangled representation learning.
> >
> > I'm not quite sure such a blanket statement can be made about IB and representation learning, particularly on the back of prior approaches to leverage IB (and the rate-distortion principle more generally) to effect meaningful representations [E.g. 1-4].
> >
> > [1] M. Vera, P. Piantanida and L. R. Vega, "The Role of the Information Bottleneck in Representation Learning," In IEEE International Symposium on Information Theory (ISIT), pp. 1580-1584, 2018.
> >
> > [2] Yamada, M., Kim, H., Miyoshi, K., Iwata, T., and Yamakawa, H. "Disentangled Representations for Sequence Data using Information Bottleneck Principle." In Proceedings of The 12th Asian Conference on Machine Learning, in Proceedings of Machine Learning Research 129:305-320. 2020.
> >
> > [3] Jeon, Insu, Wonkwang Lee, Myeongjang Pyeon, and Gunhee Kim. "Ib-gan: Disentangled representation learning with information bottleneck generative adversarial networks." In Proceedings of the AAAI conference on artificial intelligence, vol. 35, no. 9, pp. 7926-7934. 2021.
> >
> > [4] Gege Gao, Huaibo Huang, Chaoyou Fu, Zhaoyang Li, Ran He; "Information Bottleneck Disentanglement for Identity Swapping" Proceedings of the IEEE/CVF Conference on Computer Vision and Pattern Recognition (CVPR), pp. 3404-3413, 2021.
> >
> > > [...]  only limitation of our metrics is the way the regressors are chosen [...] also make use of the regressors
> >
> > Right, its reasonable that there is some constraint on the complexity of the regressors, but what is not clear is that the sensitivity to what type of regressor used is the same in these prior approaches?
> >
> > Overall, I'm still unconvinced that this work, as presented, covers the connection to prior work sufficiently to understand exactly what benefits it brings. On this evidence, I will stick with my original score, although I do appreciate the effort the authors have taken to provide a rebuttal.

---

> ### Author Response · Authors · 2024-11-25
> **Thank you for your response**
>
> Thank you again for your feedback. We answer below your last comments.
>
> > I'm a bit surprised by this; generally, for the less 'extensive' datasets to explore disentanglement in, CelebA is known to be the case where one cannot assume independence between factors (e.g. beard vs gender, etc). It would be useful to get a handle on how much correlation would be required for this method to be useful then?
>
> You are right. There is some correlation between factors in CelebA. However, what we meant is that, since many of the factors are almost uncorrelated to each other, the total correlation is very low. This implies that other metrics approximately perform reasonably well. Since we agree in that analyzing the case with more factors of variation is interesting, we added a similar experiment to that of Section 5.2 but for CelebA. Thus, we can analyze what happens when we have more factors of variation and strong correlations between factors. You can find the results and description in Appendices D and E, respectively.
>
> > I'm not quite sure such a blanket statement can be made about IB and representation learning, particularly on the back of prior approaches to leverage IB (and the rate-distortion principle more generally) to effect meaningful representations.
>
> You are right. We weren't aware of those papers. The paragraph related to the IB in section 2 has been appropriately modified. The main difference between our works and the previous is that ours formally define a definition of disentanglement, as stated now in the paper.
>
> We hope that these two last changes have helped to improve the level of convincingness that you believe our work has. You pointed out two weaknesses in the original review and we have tried to do our best to fix them:
> 1. The absence of experimentation with more realistic datasets, such as CelebA. We performed some experiments with the original CelebA and adding some extra correlations between factors of variation of this.
> 2. The absence of mention and connection of our work to the Information Bottleneck in section 2. We addressed this: Now we introduce the concept of IB and we connect it to that of disentangled representation learning. We specify which is the main novelty of our work with the other works connecting IB and disentangled representation learning.
> If you are still unconvinced, we would appreciate if you could be more precise on what you think that our work is missing to result more convincing.
>
> > Right, its reasonable that there is some constraint on the complexity of the regressors, but what is not clear is that the sensitivity to what type of regressor used is the same in these prior approaches?
>
> We would include in the final version some figures (similar to those in sections 5.1 and 5.2) comparing different regressors for our metrics. This would help to show the robustness of them.

---

> > ### Author Response · Authors · 2024-11-30
> > **Inclusion of more complex representations**
> >
> > Thank you again for your review. We have performed some experiments including neural representations with higher dimensions and more complex relation with the factors of variation.
> >
> > We hope that this served to improve your opinion on the paper. If you believe that it deserves a score raise, we would appreciate if you made that raise effective.

---

### Official Review · Reviewer_nQpi · 2024-11-07

**Soundness:** 3
**Presentation:** 3
**Contribution:** 3
**Rating:** 8
**Confidence:** 3

**Summary:**

This paper explores disentangled representation learning, where the goal is to separate factors of variation in data into understandable parts. Traditional definitions assume these factors are independent, which is often unrealistic in real-world data, limiting their applicability. The authors introduce a new, information-theory-based definition of disentanglement that works even when factors are not independent. They show that this new definition aligns with representations made up of minimal and sufficient variables. Additionally, they propose a method to measure disentanglement in these scenarios and demonstrate its effectiveness in handling both independent and non-independent factors, outperforming existing metrics.

**Strengths:**

- The paper provides a great theoretical framework for understanding the nature of disentanglement. It presents a set of succinct conditions that are essential to disentanglement and condenses them into two highly general notions: minimality and sufficiency.
- The proposed disentanglement measure does not require independent latent factors, addressing a key limitation of previous approaches.
- The paper is very well written. It includes an extensive discussion of the related work and a nicely listed discussion on the desirable properties of a disentangled representation. The whole derivation process from the basic properties to the final algorithm is very clear and easy to follow.

**Weaknesses:**

- The proposed metrics (minimality and sufficiency) may not be very informative in cases where $y$ can’t be perfectly recovered from $x$. In such cases, no representation of $x$ can achieve 100% minimality and sufficiency, meaning that the factors can’t be perfectly disentangled. As a result, the optimal value of minimality/sufficiency may be different for different tasks. In general, this value is a priori unknown and may not be easy to estimate. This makes it difficult to tell if a representation is good or bad (in terms of disentanglement) if the measurement of minimality/sufficiency yields a medium value (not close to 0 or 1).
- The measurement algorithms require the ground-truth values of the causal factors $y$, which may be difficult to obtain for many real-world tasks. Moreover, in cases where $y$ has to be estimated from $x$, the estimation quality could affect the measurement of minimality/sufficiency.
- The metrics are defined as ratios between two mutual information terms in order to scale them between 0 and 1, but is this really a good choice? For example, do $\frac{I(z_j;y_i)}{I(z_j;x)}=\frac{0.01}{0.1}$ and $\frac{I(z_j;y_i)}{I(z_j;x)}=\frac{10}{100}$ really mean the same thing for minimality? Note that the values of $I(z_j;x|y_i)$ are quite different in these two cases. The definition deserves a more careful discussion.
It seems that the experiments only involve problems with a small number of causal factors. Is the proposed measure also accurate in much higher-dimensional spaces? Will there be computational issues? The time complexity of the measurement algorithm is O(mn).
- Why are the X, Y, and Z in the first paragraph of section 4 in capitals whereas the rest of the paper uses lower cases?
- Typos: 1. Line 69-70: missing “of” between “degree” and “minimality”; 2. Line 229: have *the* that; 3. Line 4 of Algorithm 4: n should be m.

**Questions:**

Please see the weaknesses.

---

> ### Author Response · Authors · 2024-11-23
> **Answer to Reviewer nQPi**
>
> We would like to start by thanking you for your review. Next we try to address each of your questions and concerns. Next bullet points follow the same order as yours in Weaknesses section:
>
> * We consider that the factors of variation can be always completely defined by $x$. As it has been noted in identifiability literature, if the mixing function that generates $x$ from the factors of variation $y$ and nuisances $n$ is not invertible, then the problem is ill-posed (see Appendix A.5 in [1]). The fact that the this mixing function is considered to be invertible implies that all the information of all the factors of variation of $y$ is present in $x$. Thus, the best and worst possible values of minimality/sufficiency are always 1 and 0, respectively.
> * It is true that the ground-truth values of $y$ need to be known in order to calculate the minimality/sufficiency, but as far as we know, this is a "problem" that all the metrics in the literature (those described in section 2) suffer.
>     We could think of a scenario in which there is a subset $y'^{(k)} \subseteq y^{(k)}$ whose ground-truth value is known for a given input $x^{(k)}$, similarly to [2]. In this case, line 4 of Algorithm 1 and line 3 of Algorithm 2 should not be $i=1$ to $n$, but $i \in y'^{(k)}$. We believe, however that maybe this can be more confusing for the reader and we assume that all the ground-truth values are known, as other metrics do.
> * The fact that the metrics are normalized is intentional and does not suppose any loose of robustness (in fact, the lack of robustness would come from having unnormalized quantities). We detail this next:
>     Since $z_j$ is a representation of $x$, we know that $H(z_j|x)=0$ and, thus $H(z_j)=I(z_j;x)$. This means that in both scenarios that you propose $y_i$ "explains" a $10\%$ of $z_j$ (more correctly, contains a $10\%$ of $z_j$ of the information of $y_i$). Thus, the interpretation of minimality is the proportion of information of $z_j$ that $y_i$ contains with respect to $x$ in all the cases, no matter how entropic $z_j$ is. This is because some variables can be more entropic than others, but minimality should be robust to this. Equivalent reasoning can be applied to Sufficiency.
>     From a theoretical point of view, we should not find any problem when having more factors and the time complexity is not higher than that of other algorithms in the literature.
> * Lowercase notation is used in all the paper now.
> * Typos have been solved. Thanks for noticing.
>
> We hope that we have clarified your questions and strengthen our weaknesses. If so and you believe that current version of the work deserves an increase in the score, we would appreciate it if you make that increase effective. Otherwise, we will be willing to continue with the discussion.
>
> [1] Lachapelle, S., Rodriguez, P., Sharma, Y., Everett, K. E., Le Priol, R., Lacoste, A., \& Lacoste-Julien, S. (2022, June). Disentanglement via mechanism sparsity regularization: A new principle for nonlinear ICA. In Conference on Causal Learning and Reasoning (pp. 428-484). PMLR.
>
> [2] Locatello, F., Abbati, G., Rainforth, T., Bauer, S., Schölkopf, B., \& Bachem, O. (2019). On the fairness of disentangled representations. Advances in neural information processing systems, 32.

---

> ### Comment · Reviewer_nQpi · 2024-11-26
>
> Thank you for the clarifications. Below are my further comments regarding the first point.
>
> > We consider that the factors of variation can be always completely defined by $x$.
>
> Yes, I understand that. However, I don't think the non-invertible setting is ill-posed, as many real-world problems are indeed not invertible. Appendix A.5 in [1] does not say the problem is ill-posed either. My point is that one may not know a priori how invertible a given problem is, and if it is not invertible, then proposed metrics may not reflect how good a representation really is. I agree with Reviewer M4R5 that properly ranking different representations may be a more important utility of the metrics here. On the other hand, the absolute score is only useful when the maximum possible score for the problem is known (e.g., about 100% for metrics like accuracy on most datasets).

---

### Author Response · Authors · 2024-11-23
**Answer to All Reviewers**

First, we want to thank to all the reviewers for their feedback. We appreciate all the comments and we honestly believe that they have helped to improve the quality of the work. One of the parts that received more criticism is Experiment 5.2 and we agree that this did not reflect clearly the advantage of our metrics over others in the literature. Thus, we have modified this a little bit. You can find it in the new version of the manuscript. The main changes with respect to the previous version of the experiment are listed below:

* In the previous version of Experiment 5.2, only a pair of factors were correlated, which translated into a low level of total correlation between factors. Thus, other metrics were performing reasonably well in this case, which did not allow our metric to "shine", as Reviewer M4R5 pointed out. Thus, similar to what [1] does, we introduce correlations between different number of pairs of data. We believe that this allows to have a better view on how the level of correlation between factors negatively affect to other metrics even in cases of perfect disentanglement.
* A deeper explanation of the experiment is intended to be given. Since the results under a higher level of correlation between factors are clearer to interpret, we believe that this also translates into a  clearer explanation.
* As Reviewer k1fY pointed out, no standard deviations were given in this experiment. We added them in Figures 5, 6, 10-13.

All the comments referring to different topics are answered one by one to each reviewer.


[1] Roth, K., Ibrahim, M., Akata, Z., Vincent, P., \& Bouchacourt, D. (2022). Disentanglement of correlated factors via hausdorff factorized support. arXiv preprint arXiv:2210.07347.

---

### Author Response · Authors · 2024-11-30
**Modification of Experiment 5.2. Neural representations and correlation to downstream tasks I.**

Thanks to all of you for your time and answers. Since it was the part that you reccomend us the most, we have been thinking in (i) different ways of showing why metrics that consider correlated factors of variation are important in different downstream tasks and (ii) how our metrics perform with actual neural representation. For that purpose, we have modified section 5.2 and we have trained a set of neural encoders (all of them from VAEs variation) with the correlated varsions of Shapes3d. If the paper is finally accepted, we will include experiments with MPI3D and Dsprites. We had no time during this week to obtain results with these datasets. We detail below the description and conclusions of the performed experiments. The results shown in tables would be in figures in the final paper, but we cannot modify the file now.

### 5.2. Performance in Neural Representations
As mentioned, the factors in most of the datasets in the literature for measuring disentanglement are independent.
However, [1] proposes a modification to introduce correlation between pairs of factors as $p(y_1,y_2) \propto \exp\left(-(y_1-\alpha y_2)^2/2\sigma^2\right)$, where $\alpha = c_2^{\max}/c_2^{\max}$, i.e., the lower the value of $\sigma$ the stronger the correlation. Subsequently, [2] proposes to introduce this correlation scheme between multiple pairs of factors. We make use of this scheme to compare the metrics of previous section in the neural representations, as we explain below:

First, we train a set representations extractors whose outputs are intended to be disentangled. These are encoders of different VAE variations (VAE, $\beta$-VAE, $\beta$-TCVAE, FactorVAE, AnnealedVAE and AdaGVAE) and they are trained only with uncorrelated data.
Second, we define a set of scenarios, where each of them is fully defined by the distribution of the factors of variation, i.e., different scenarios have different distribution of $p(y)$ and this is the only difference between different scenarios. Specifically, $p(y)$ is defined by using the formula of the previous paragraph for all the scenarios.
Each scenario is considered to have a level of correlation, which corresponds to the number of correlated pairs. Concretely, we have scenarios with uncorrelated factors (original dataset), correlations between multiple pairs of factors (a factor is never correlated with more than one factor) and shared confounders (one factor correlates to all others). The level of correlation could be also defined through $\sigma$, but we set it constant to $0.1$ in all the cases for simplicity.
Finally, we obtain all the metrics described in the previous section for every representations extractor and scenarios. Next we analyze how these metrics vary depending on the level of correlation of the factors.

**Reconstruction Error and Minimality Metrics**

Metrics to measure minimality aim to capture if each variable of the representation is affected by a single factor regardless of the others. Thus, a representation $z^{(k)}$ corresponding to the input $x^{(k)}$ should have always the same value of minimality independently of the rest of the elements of the dataset, just as $z^{(k)}$ results in the same quality reconstruction regardless of the rest of the elements of the dataset. Thus, although minimality and reconstruction error are not necessarily correlated [3], the rank correlation between them should hold along the different levels of correlation. In other words, those elements that have high minimality in an uncorrelated case should have a high minimality in a correlated case, and vice versa. Since this holds also for the reconstruction error, we analyze the Spearman rank correlation between the different minimality metrics and reconstruction error, which is shown in Figure 5. In this, we can see that _Minimality_ is the only metric whose rank correlation with the reconstruction error remains almost constant for all the factors correlation levels. This indicates that the rest of the metrics are highly dependent on the correlation level of the factors of variation.


**Sample Efficiency and Sufficiency Metrics**

As explained in different works [2,3,4], a disentangled representation should translate into a more sample efficient downstream predictor. Intuitively, the predictor should focus only on a few variables of the representation to make a prediction, so only a few parameters of the predictor are important. Concretely, metrics to measure sufficiency should correlate to the sample efficiency. We define sample efficiency as the average accuracy based on 100 samples divided by the average accuracy based on 10 000 samples. First, we can see that all the metrics except MIG are highly correlated between them and with sample efficiency in the uncorrelated case. However, while the level of correlation increases, _Sufficiency_ is the only one that holds this high correlation.

---

> ### Author Response · Authors · 2024-11-30
> **Tables**
>
> Fig 5.a. Uncorrelated
> |             | $\beta$-VAE | Factor-VAE | DCI D | Modularity | Minimality | Rec. Error |
> |-------------|-------------|------------|-------|------------|------------|------------|
> | $\beta$-VAE | 100 | 100 | 73 | 21 | 73 | -94 |
> | Factor-VAE | 100 | 100 | 73 | 21 | 73 | -94 |
> | DCI D | 73 | 73 | 100 | 80 | 100 | -60 |
> | Modularity | 21 | 21 | 80 | 100 | 80 | 0 |
> | Minimality | 73 | 73 | 100 | 80 | 100 | -60 |
> | Rec. Error | -94 | -94 | -60 | 0 | -60 | 100 |
>
> Fig 5.b. 1 Pair
> |             | $\beta$-VAE | Factor-VAE | DCI D | Modularity | Minimality | Rec. Error |
> |-------------|-------------|------------|-------|------------|------------|------------|
> | $\beta$-VAE | 100 | 68 | 38 | 44 | 70 | -63 |
> | Factor-VAE | 68 | 100 | 61 | 41 | 64 | -47 |
> | DCI D | 38 | 61 | 99 | 35 | 46 | -16 |
> | Modularity | 44 | 41 | 35 | 99 | 80 | -49 |
> | Minimality | 70 | 64 | 46 | 80 | 99 | -61 |
> | Rec. Error | -63 | -47 | -16 | -49 | -61 | 100 |
>
> Fig 5.c. 2 Pairs
> |             | $\beta$-VAE | Factor-VAE | DCI D | Modularity | Minimality | Rec. Error |
> |-------------|-------------|------------|-------|------------|------------|------------|
> | $\beta$-VAE | 100 | 65 | 19 | 47 | 56 | -49 |
> | Factor-VAE | 65 | 100 | 50 | 52 | 61 | -47 |
> | DCI D | 19 | 50 | 100 | 27 | 39 | -1 |
> | Modularity | 47 | 52 | 27 | 99 | 85 | -57 |
> | Minimality | 56 | 61 | 39 | 85 | 100 | -59 |
> | Rec. Error | -49 | -47 | -1 | -57 | -59 | 100 |
>
> Fig 5.d. 3 Pairs
> |             | $\beta$-VAE | Factor-VAE | DCI D | Modularity | Minimality | Rec. Error |
> |-------------|-------------|------------|-------|------------|------------|------------|
> | $\beta$-VAE | 100 | 66 | 11 | 49 | 51 | -39 |
> | Factor-VAE | 66 | 99 | 46 | 60 | 61 | -48 |
> | DCI D | 11 | 46 | 99 | 17 | 30 | 14 |
> | Modularity | 49 | 60 | 17 | 99 | 86 | -61 |
> | Minimality | 51 | 61 | 30 | 86 | 99 | -57 |
> | Rec. Error | -39 | -48 | 14 | -61 | -57 | 100 |
>
> Fig 5.e. Confounded
> |             | $\beta$-VAE | Factor-VAE | DCI D | Modularity | Minimality | Rec. Error |
> |-------------|-------------|------------|-------|------------|------------|------------|
> | $\beta$-VAE | 100 | 82 | 31 | 83 | 74 | -74 |
> | Factor-VAE | 82 | 100 | 53 | 80 | 76 | -62 |
> | DCI D | 31 | 53 | 100 | 47 | 46 | -11 |
> | Modularity | 83 | 80 | 47 | 100 | 90 | -67 |
> | Minimality | 74 | 76 | 46 | 90 | 100 | -62 |
> | Rec. Error | -74 | -62 | -11 | -67 | -62 | 99 |
>
> Fig 6.a. Uncorrelated
> |             | MIG | SAP | DCI C | Explicitness | Sufficiency | Efficiency |
> |-------------|-------------|------------|-------|------------|------------|------------|
> | MIG | 100 | 40 | 20 | 40 | 20 | 20 |
> | SAP | 40 | 100 | 80 | 100 | 80 | 80 |
> | DCI C | 20 | 80 | 100 | 80 | 100 | 100 |
> | Explicitness | 40 | 100 | 80 | 100 | 80 | 80 |
> | Sufficiency | 20 | 80 | 100 | 80 | 100 | 100 |
> | Efficiency | 20 | 80 | 100 | 80 | 100 | 100 |
>
> Fig 6.b. 1 Pair
> |             | MIG | SAP | DCI C | Explicitness | Sufficiency | Efficiency |
> |-------------|-------------|------------|-------|------------|------------|------------|
> | MIG | 99 | 72 | 72 | 11 | -21 | -16 |
> | SAP | 72 | 99 | 68 | 53 | 23 | 26 |
> | DCI C | 72 | 68 | 99 | 27 | 25 | 24 |
> | Explicitness | 11 | 53 | 27 | 99 | 59 | 58 |
> | Sufficiency | -21 | 23 | 25 | 59 | 99 | 86 |
> | Efficiency | -16 | 26 | 24 | 58 | 86 | 99 |
>
> Fig 6.b. 2 Pairs
> |             | MIG | SAP | DCI C | Explicitness | Sufficiency | Efficiency |
> |-------------|-------------|------------|-------|------------|------------|------------|
> | MIG | 100 | 66 | 75 | -6 | -20 | -13 |
> | SAP | 66 | 99 | 53 | 27 | 8 | 11 |
> | DCI C | 75 | 53 | 100 | 11 | 17 | 20 |
> | Explicitness | -6 | 27 | 11 | 100 | 62 | 57 |
> | Sufficiency | -20 | 8 | 17 | 62 | 100 | 88 |
> | Efficiency | -13 | 11 | 20 | 57 | 88 | 100 |
>
> Fig 6.c. 3 Pairs
> |             | MIG | SAP | DCI C | Explicitness | Sufficiency | Efficiency |
> |-------------|-------------|------------|-------|------------|------------|------------|
> | MIG | 99 | 58 | 78 | -36 | -19 | -17 |
> | SAP | 58 | 100 | 41 | -9 | -5 | -2 |
> | DCI C | 78 | 41 | 99 | -24 | 14 | 11 |
> | Explicitness | -36 | -9 | -24 | 99 | 67 | 64 |
> | Sufficiency | -19 | -5 | 14 | 67 | 99 | 87 |
> | Efficiency | -17 | -2 | 11 | 64 | 87 | 99 |
>
> Fig 6.a. Confounded
> |             | MIG | SAP | DCI C | Explicitness | Sufficiency | Efficiency |
> |-------------|-------------|------------|-------|------------|------------|------------|
> | MIG | 100 | 79 | 78 | -1 | -14 | 0 |
> | SAP | 79 | 100 | 63 | 34 | 7 | 21 |
> | DCI C | 78 | 63 | 100 | 12 | 28 | 35 |
> | Explicitness | -1 | 34 | 12 | 99 | 57 | 60 |
> | Sufficiency | -14 | 7 | 28 | 57 | 100 | 81 |
> | Efficiency | 0 | 21 | 35 | 60 | 81 | 100 |

---

> ### Author Response · Authors · 2024-11-30
> **References**
>
> [1] Träuble, F., Creager, E., Kilbertus, N., Locatello, F., Dittadi, A., Goyal, A., ... & Bauer, S. (2021, July). On disentangled representations learned from correlated data. In International conference on machine learning (pp. 10401-10412). PMLR.
>
> [2] Roth, K., Ibrahim, M., Akata, Z., Vincent, P., & Bouchacourt, D. (2022). Disentanglement of correlated factors via hausdorff factorized support. arXiv preprint arXiv:2210.07347.
>
> [3] Locatello, F., Bauer, S., Lucic, M., Raetsch, G., Gelly, S., Schölkopf, B., & Bachem, O. (2019, May). Challenging common assumptions in the unsupervised learning of disentangled representations. In international conference on machine learning (pp. 4114-4124). PMLR.
>
> [4] Ng, A. Y. (2004, July). Feature selection, L 1 vs. L 2 regularization, and rotational invariance. In Proceedings of the twenty-first international conference on Machine learning (p. 78).

---

### Meta-Review · Area_Chair_pr6S · 2024-12-22

**Metareview:**

The paper presents a theoretically sound approach to defining disentanglement metrics that are more applicable to real-world scenarios involving correlated generative factors and nuisance variables. The introduction is well-written and effectively motivates the research by clearly laying out the gaps in existing metrics and their practical relevance. The background and theory are systematically developed, making the paper easy to follow and serving as a valuable resource for the community. The proposed properties—factors-invariance, nuisances-invariance, representations-invariance, and explicitness—are well-argued and extend existing notions. The theoretical link between minimality and sufficiency and the proposed metrics is convincing. The work also demonstrates the metrics’ utility through illustrative experiments, showcasing their potential to complement traditional disentanglement metrics.

Despite the strong theoretical contributions, the empirical validation is narrow and less compelling. The experiments lack sufficient exploration of practical downstream utility, such as model selection for tasks like generalization, fairness, or sample efficiency. In correlated scenarios, where the new metrics should excel, they sometimes perform suboptimally compared to traditional metrics like DCI. The absence of experiments on learned representations, even in controlled scenarios, limits the applicability of the findings to realistic settings. The choice of datasets and the exclusion of image data usage raises questions about experimental design. Lastly, the claims about the practical superiority of the metrics, especially in uncorrelated cases, seem overstated given that traditional metrics already perform well in such scenarios. These shortcomings highlight a need for more comprehensive experiments and clearer framing of the work’s practical implications.

**Additional Comments On Reviewer Discussion:**

Great discussion, reviewer M4R5 did a great job.

---

### Decision · Program_Chairs · 2025-01-22

Reject